# Incremental Feature Selection in Dynamic Incomplete Ordered Decision Systems

## Abstract

Incremental feature selection aims to efficiently identify key features from dynamic data. However, existing feature selection algorithms for dynamic incomplete ordered data often rely on upper and lower approximations while overlooking the impact of inter-feature relationships across different decision classes. This can lead to reduced computational efficiency and suboptimal classification accuracy. To address these issues, this paper proposes an incremental feature selection method based on expanded dominance matrices for incomplete ordered decision systems. Firstly, we propose to use non-dominant relationships between classes as a measure of attribute importance, thereby avoiding the computational complexity of traditional lower and upper approximation. Furthermore, to maintain efficiency and accuracy in dynamic data environments which involve frequent object addition and deletion, we propose two matrix-based incremental update mechanisms: matrix-based non-dominance attribute reduction for addition (MNAR-A) and matrix-based non-dominance attribute reduction for deletion (MNAR-D). These mechanisms are crucial for efficiently updating the feature subset when new objects are added or existing objects are removed, ensuring the algorithm remains effective and avoids recomputing from scratch. Experimental results on the UCI dataset showed that the proposed algorithm achieved a $1.3\times$ speedup and delivered a 7% relative accuracy gain compared to the state-of-the-art method on average.

## 1 Introduction

Feature selection is a commonly used data preprocessing technique in machine learning and data mining, aiming to curtail dimensionality by eliminating redundant or irrelevant attributes so as to enhance generalization and computational efficiency Wang et al. (2024); Ding et al. (2021); Chen et al. (2020). However, real-world data often exhibit large-scale and dynamically evolving characteristics (*e.g.*, real-time data streams and continuously updated databases).

Conventional static feature selection methods are not suitable for dynamically evolving data, as they must recompute reducts from scratch, incurring substantial redundancy and computational overhead. Incremental learning approaches Luo et al. (2014); Zhang et al. (2025); Zhao et al. (2024b) offer a promising solution by effectively leveraging previously obtained results and knowledge for efficient feature selection in dynamic data environments. Meanwhile, rough set theory Pawlak (1982) and its extensions:particularly the dominance-based rough set approach Greco et al. (2002; 2001), provide a solid theoretical foundation for handling ordered data endowed with preference ranking information (*e.g.*, ratings and scores).

However, the widespread presence of missing or incomplete data in practical applications, such as fault diagnosis Ge et al. (2018) and uncertainty assessment Dai et al. (2017); Feng & Jing (2016), make it an urgent and open research problem to achieve efficient and accurate feature selection over dynamically evolving ordered data with missing values. Existing incremental algorithms usually focus on either ordered data or incomplete data individually, lacking effective methods to handle the dynamic changes of ordered data with missing values. Moreover, existing incremental approaches often do not adequately consider the impact of features on classification results.

To address these issues, we propose a hybrid scoring strategy. Firstly, we introduce Susmaga's concept Susmaga (2014) of inter-class non-dominance (class distinction score) to incomplete ordered decision systems (IODS) and propose a corresponding calculation method. We further propose global distinction score as an attribute importance evaluation index to enhance the accuracy of attribute reduction. We present a matrix-form incremental update mechanism for adding or deleting samples in IODS, along with the corresponding incremental attribute reduction algorithms: matrix-based non-dominance attribute reduction for addition (MNAR-A) and matrix-based non-dominance attribute reduction for deletion (MNAR-D).

The main contributions of this paper are summarized as follows.

- To capture the impact of inter-feature relationships across decision classes, we present a hybrid scoring strategy. We incorporate Susmaga's inter-class non-dominance (class distinction score) into IODS and further propose global distinction score as an attribute importance metric. This approach enhances the accuracy of attribute reduction.

- The proposed methods accelerate feature selection in IODS, yielding a moderate speedup by harnessing the high computational efficiency of matrix operations for incremental calculation.

## 2 Related work

This paper primarily focuses on the issue of "missing values" in ordered information systems and studies attribute reduction based on expanded dominance relations in such datasets. To cope with the pervasive missing values in practice, researchers have proposed expanded method Shao & Zhang (2005); Guan et al. (2018), similarity-based method Yang et al. (2008), and feature-based dominance relations method Du & Hu (2016).

Constructing an effective attribute evaluation function is central to attribute reduction. Typical measures —*e.g.*, dependency Ullah et al. (2024), entropy Sang et al. (2021); Xu et al. (2024b), and knowledge granularity Jing et al. (2016); Liu & Feng (2022)— mainly describe intrinsic attribute characteristics and overlook their direct impact on classification. To bridge this gap, Hu et al. (2021) introduced a separability metric for fuzzy decision systems, while Susmaga defined intra- and inter-class reducts via proximity and non-dominance relations in ordered data Susmaga (2014). However, these approaches assume static datasets; when data evolve continuously, reducts must be recomputed from scratch, leading to poor efficiency. We adopt incremental computation to substantially reduce the computational overhead incurred by data changes.

Incremental computation uses existing knowledge to improve the efficiency of knowledge discovery Ding et al. (2022). To address dynamic changes in samples within incomplete information systems, Zhao et al. (2024a) employed sub-tolerance relations to reduce runtime. For dynamic changes in samples within ordered information systems, Xu et al. (2024a) applied conditional entropy to attribute reduction. Existing incremental algorithms seldom address ordered and incomplete data simultaneously and often overlook the influence of features on classification. By introducing a hybrid scoring strategy, our method not only improves attribute reduction accuracy but also mitigates the impact of missing values and handles ordered data more effectively.

## 3 Preliminaries

An information system Pawlak (1992) is typically defined as a 4-tuple $S = (U, A, V, f)$, where $U = \{x_1, x_2, \cdots x_n\}$ is a non-empty finite set of objects, called the universe. $A$ denotes the set of all attributes. In decision system, $A = C \cup D$ and $C \cap D = \emptyset$, where $C$ is a non-empty finite set of condition attributes and $D$ is decision attribute. $V$ is regarded as the domain of all attributes. $f : U \times A \to V$ is an information function such that $f(x, a) \in V_a, \forall a \in A$ and $x \in U$, where $V_a$ is the domain of attribute $a$.

### 3.1 Ordered information system and dominance relation

If each attribute domain $V_a$ is equipped with a preference preorder $\succeq_a$, then $a$ is called a criterion attribute. The preorder satisfies reflexivity and transitivity. When all attributes in the system are criterion attributes, the system is called an ordered information system (OIS), denoted by $S^{\succeq} = (U, A, V, f)$.

For a subset of condition attributes $P \subseteq C$, the dominance relation $D_P$ on $U$ is defined as follows: for any $x, y \in U$,

$$xD_Py \iff \forall a \in P, f(x,a) \succeq_a f(y,a). \tag{1}$$

Correspondingly, the $P$-dominating and $P$-dominated sets of $x \in U$ are defined as

$$D_P^+(x) = \{y \in U \mid yD_Px\}, \quad D_P^-(x) = \{y \in U \mid xD_Py\}. \tag{2}$$

### 3.2   Incomplete ordered decision system and expanded dominance relation

In practice, condition attributes may contain missing values denoted by "$*$", while decision attributes are assumed complete. Such systems are called incomplete ordered decision systems (IODS), denoted as $S^{\succeq} = (U, C \cup \{d\}, V, f)$, where $d$ is the decision attribute.

In this situation, $D_P$ should be accommodated by unknowns and extended to $D_P^*$ : for any $(x, y) \in U \times U$

$$xD_P^*y \iff \forall a \in P, f(x,a) \succeq_a f(y,a) \ \lor f(x,a) = * \lor f(y,a) = *. \tag{3}$$

The relation $D_P^*$ is reflexive but may not be antisymmetric or transitive.

The corresponding expanded dominating and dominated sets are

$$D_P^+(x) = \{y \in U \mid yD_Px\}, \quad D_P^-(x) = \{y \in U \mid xD_Py\}. \tag{4}$$

## 4   Class distinction score in IODS

### 4.1   Set-based calculation of class distinction score

In this subsection, we present a set-based formalism for calculating the class distinction score (CDS) measure in IODS.

Based on the decision attribute $d$, the universe $U$ is partitioned into decision classes: $CL = \{cl_1, cl_2, \ldots, cl_T\}$, which are ordered by preference$cl_1 \prec cl_2 \prec \cdots \prec cl_T$, meaning $cl_i$ has a lower preference level than $cl_j$ if $i < j$.

**Definition 1** (Inter-class pairs)**.** *Define the set of inter-class pairs as:*

$$Inter\_cl = \{(cl_i, cl_j) \in CL \times CL \mid 1 \leq i < j \leq T\}. \tag{5}$$

*To ensure the dominant relationship between decision classes, the inter-class structure requires the selection of a lower class $cl_i$ and a higher class $cl_j$.*

**Definition 2** (Class interference metric (CIM))**.** *For any attribute subset $B \subseteq C$, the Class Interference Metric between two decision classes $cl_i$ and $cl_j$ is defined as the degree of dominance of $cl_j$ over $cl_i$ with respect to $B$:*

$$CIM_B(cl_i, cl_j) = \frac{\sum_{k=1}^{|cl_i|} \left| D_B^{*+}(cl_{ik}) - \overline{(cl_i, cl_j)} \right|}{(|cl_i| + |cl_j|)^2} + \frac{\sum_{k=1}^{|cl_j|} \left| D_B^{*+}(cl_{jk}) - \overline{(cl_i, cl_j)} \right|}{(|cl_i| + |cl_j|)^2}. \tag{6}$$

*Here $cl_{ik}$ represents the k-th object in the decision class $cl_i$, $|\bullet|$ represents the basis of the set (i.e., the number of elements in the set), and $\overline{(cl_i, cl_j)}$ represents the set of elements that are not in the inter-class $(cl_i, cl_j)$.*

**Definition 3** (Class distinction score (CDS)). *The Class Distinction Score of $cl_i$ and $cl_j$ with respect to the subset of attributes $B \subseteq C$ is defined as*

$$CDS_B(cl_i, cl_j) = 1 - CIM_B(cl_i, cl_j). \tag{7}$$

*A higher CDS value indicates stronger non-dominance and better classification separability between classes under attribute subset B.*

**Definition 4** (Global distinction score (GDS)). *Given $n$ Class Distinction Score values denoted as $CDS_1, CDS_2, \ldots, CDS_n$. Then the Global Distinction Score of the object set is defined as*

$$GDS_U(B, CL) = \frac{1}{n} \sum_{i=1}^{n} CDS_i. \tag{8}$$

*A higher $GDS_U(B, CL)$ value suggests that the attribute subset B provides better-ranking information and classification accuracy for the entire decision system.*

Based on the *GDS* measure, the importance of attributes can be definite as follows.

**Definition 5** (Inner significance). *Let $S^{\succeq} = (U, C \cup \{d\}, V, f)$ be an IODS $\forall B \subseteq C$ and $\forall a \in B$ the GDS-based inner significance measure of $a$ in $B$ is defined as*

$$sig_{inner}^{\succeq U}(a, B, d) = GDS_U(B, CL) - GDS_U(B \setminus \{a\}, CL) \tag{9}$$

Based on the explanation of GDS, a greater value of $sig_{inner}^{\succeq U}(a, B, d)$ implies that the conditional attribute is more crucial. This measure helps to identify the necessary condition attributes within the entire set of condition attributes. Additionally, the core attribute set of the attribute set $B$ is defined as $Core_B = \{a \in B \mid sig_{inner}^{\succeq U}(a, B, d) > 0\}$.

**Definition 6** (Outer significance). *Let $S^{\succeq} = (U, C \cup \{d\}, V, f)$ be an IODS, $\forall P \subseteq C$ and $\forall a \in (C \setminus P)$ the GDS-based outer significance measure of $a$ to $P$ is defined as*

$$sig_{outer}^{\succeq U}(a, P, d) = GDS_U(P \cup \{a\}, CL) - GDS_U(P, CL) \tag{10}$$

Similar to the $sig_{inter}^{\succeq U}(a, P, d)$, the $sig_{outer}^{\succeq U}(a, P, d)$ can be used to identify necessary condition attributes that are distinct from those in the selected condition attribute set.

**Proposition 1.** *Let $S^{\succeq} = (U, C \cup \{d\}, V, f)$, $\forall B \subseteq C$ and $B \neq \emptyset$, then $GDS_U(C, CL) \geqslant GDS_U(B, CL)$.*

*Proof.* Let $S^{\succeq} = (U, C \cup \{d\}, V, f)$, $C = \{a_1, a_2, \ldots, a_n\}$. For all $B \subseteq C$ and $X \subseteq U$, we have $D_C^{*+}(X) \subseteq D_B^{*+}(X)$, so $|D_C^{*+}(cl_{ik})| \leq |D_B^{*+}(cl_{ik})|$. According to Definition 2,

$$CIM_B(cl_i, cl_j) = \frac{\sum_{k=1}^{|cl_i|} \left| D_B^{*+}(cl_{ik}) - \overline{(cl_i, cl_j)} \right|}{(|cl_i| + |cl_j|)^2} + \frac{\sum_{k=1}^{|cl_j|} \left| D_B^{*+}(cl_{jk}) - \overline{(cl_i, cl_j)} \right|}{(|cl_i| + |cl_j|)^2}$$

we can get $CIM_C(cl_i, cl_j) \leq CIM_B(cl_i, cl_j)$. Then, according to Definition 3, $CDS_B(cl_i, cl_j) = 1 - CIM_B(cl_i, cl_j)$, averaging over all class pairs, we obtain $GDS_U(C, CL) \geq GDS_U(B, CL)$. □

## 4.2 Matrix-based calculation of class distinction score

In this subsection, we present a matrix-based formalism for calculating the Class Distinction Score measure (CDS) in IODS. This matrix representation simplifies computations and facilitates incremental updates.

**Definition 7** (Expanded dominance matrix). *Let $S^{\succeq} = (U, C \cup \{d\}, V, f)$ be an IODS with universe $U = \{x_1, x_2, \ldots, x_n\}$. For any attribute subset $B \subseteq C$, the expanded dominance relation $D_B^*$ induces an $n \times n$ expanded dominance matrix on $U$, denoted by*

$$\mathbb{M}_U^{\succeq B} = [m_{(i,j)}^B]_{n \times n}, \tag{11}$$

*where the matrix entries are defined as*

$$m_{(i,j)}^B = \begin{cases} 1, & \text{if} \quad x_j D_B^* x_i, \\ 0, & \text{otherwise.} \end{cases} \tag{12}$$

**Corollary 1** (Matrix-based class interference metric (MCIM)). *For two decision classes $cl_i, cl_j \in CL$, consider their union $X = cl_i \cup cl_j$ with cardinality $m = |X|$. The matrix-based Class Interference Metric (MCIM) of attribute subset $B$ is computed as*

$$MCIM_{(cl_i, cl_j)} = \frac{\sum_{a=1}^{m} \sum_{b=1}^{m} m_{ab}^B}{m^2} = \frac{Sum(\mathbb{M}_X^{\succeq B})}{m^2}, \tag{13}$$

*where $Sum(\mathbb{M}_X^{\succeq B})$ denotes the total sum of all entries in the submatrix.*

*Proof.* According to Definition 2, we can get

$$CIM_B(cl_i, cl_j) = \frac{\sum_{k=1}^{|cl_i|} \left| D_B^{*+}(cl_{ik}) - \overline{(cl_i, cl_j)} \right|}{(|cl_i| + |cl_j|)^2} + \frac{\sum_{k=1}^{|cl_j|} \left| D_B^{*+}(cl_{jk}) - \overline{(cl_i, cl_j)} \right|}{(|cl_i| + |cl_j|)^2}.$$

According to Definition 7 and Corollary 1 , the expanded dominance relation matrix of $(cl_i, cl_j)$ is defined as: $\mathbb{M}_X^{\succeq B} = [m_{(a,b)}^B]_{m \times m}$, and

$$MCIM_{(cl_i, cl_j)} = \sum_{a=1}^{m} \sum_{b=1}^{m} \frac{m_{ab}^B}{m^2} = \frac{Sum(\mathbb{M}_X^{\succeq B})}{m^2} \tag{14}$$

where

$$m_{(a,b)}^B = \begin{cases} 1, x_b D_B^* x_a; \\ 0, otherwise. \end{cases} \tag{15}$$

, so $Sum(\mathbb{M}_X^{\succeq B}) = \sum_{k=1}^{|cl_i|} \left| D_B^{*+}(cl_{ik}) - \overline{(cl_i, cl_j)} \right| + \sum_{k=1}^{|cl_j|} \left| D_B^{*+}(cl_{jk}) - \overline{(cl_i, cl_j)} \right|$. Thus, we can get $CIM_{(cl_i, cl_j)} = MCIM_{(cl_i, cl_j)}$. In summary, the results of calculating the Class Interference Metric based on matrix and non-matrix methods are consistent. $\square$

**Corollary 2** (Matrix-based class distinction score). *The matrix-based Class Distinction Score (CDS) between $cl_i$ and $cl_j$ is given by*

$$MCDS_{(cl_i, cl_j)} = 1 - MCIM_{(cl_i, cl_j)} = 1 - \frac{Sum(\mathbb{M}_X^{\succeq B})}{m^2}. \tag{16}$$

**Corollary 3** (Matrix-based global distinction score (MGDS)). *Given all $n$ inter-class pairs $\{(cl_i, cl_j)\}$ with corresponding MCDS values $MCDS_1, MCDS_2, \ldots, MCDS_n$, the overall Class Distinction Score is defined as*

$$MGDS_U(B, CL) = \frac{1}{n} \sum_{i=1}^{n} MCDS_i. \tag{17}$$

Table 1: An example of incomplete ordered decision system.

| U | $a_1$ | $a_2$ | $a_3$ | $a_4$ | $d$ |
|---|---|---|---|---|---|
| $x_1$ | 1 | 2 | * | 1 | 1 |
| $x_2$ | 2 | 3 | 3 | 3 | 3 |
| $x_3$ | 2 | 2 | 2 | 3 | 2 |
| $x_4$ | 1 | * | 2 | 1 | 1 |
| $x_5$ | 1 | * | 2 | 2 | 2 |
| $x_6$ | 3 | 3 | 3 | 3 | 3 |
| $x_7$ | 1 | 2 | 2 | 3 | 2 |
| $x_8$ | 3 | * | 3 | 3 | 3 |
| $x_9$ | 2 | 2 | 3 | 3 | 3 |

**Example 1.** *Consider the IODS in Table 1 with attribute subset $B = \{a_1, a_2, a_3, a_4\}$ and decision classes $CL = \{cl_1, cl_2, cl_3\}$ where*

$$cl_1 = \{x_1, x_4\}, \quad cl_2 = \{x_3, x_5, x_7\}, \quad cl_3 = \{x_2, x_6, x_8, x_9\}.$$

*The inter-class pairs are $\{cl_1, cl_2\}$, $\{cl_1, cl_3\}$, and $\{cl_2, cl_3\}$. Define corresponding object sets:*

$$X_1 = cl_1 \cup cl_2 = \{x_1, x_3, x_4, x_5, x_7\}, \tag{18}$$
$$X_2 = cl_1 \cup cl_3 = \{x_1, x_2, x_4, x_6, x_8, x_9\}, \tag{19}$$
$$X_3 = cl_2 \cup cl_3 = \{x_2, x_3, x_5, x_6, x_7, x_8, x_9\}. \tag{20}$$

*Using Definition 7 the expanded dominance matrices $\mathbb{M}_{\bar{U}}^{\succeq B}$, $\mathbb{M}_{\bar{X}_1}^{\succeq B}$, $\mathbb{M}_{\bar{X}_2}^{\succeq B}$, and $\mathbb{M}_{\bar{X}_3}^{\succeq B}$ are computed as follows:*

$$\mathbb{M}_{\bar{U}}^{\succeq B} = \begin{bmatrix} 1 & 1 & 1 & 1 & 1 & 1 & 1 & 1 & 1 \\ 0 & 1 & 0 & 0 & 0 & 1 & 0 & 1 & 0 \\ 0 & 1 & 1 & 0 & 0 & 1 & 0 & 1 & 1 \\ 1 & 1 & 1 & 1 & 1 & 1 & 1 & 1 & 1 \\ 0 & 1 & 1 & 0 & 1 & 1 & 1 & 1 & 1 \\ 0 & 0 & 0 & 0 & 0 & 1 & 0 & 1 & 0 \\ 0 & 1 & 1 & 0 & 0 & 1 & 1 & 1 & 1 \\ 0 & 0 & 0 & 0 & 0 & 1 & 0 & 1 & 0 \\ 0 & 1 & 0 & 0 & 0 & 1 & 0 & 1 & 1 \end{bmatrix}_{9 \times 9},$$

$$\mathbb{M}_{\bar{X}_1}^{\succeq B} = \begin{bmatrix} 1 & 1 & 1 & 1 & 1 \\ 0 & 1 & 0 & 0 & 0 \\ 1 & 1 & 1 & 1 & 1 \\ 0 & 1 & 0 & 1 & 1 \\ 0 & 1 & 0 & 0 & 1 \end{bmatrix}_{5 \times 5}, \quad \mathbb{M}_{\bar{X}_2}^{\succeq B} = \begin{bmatrix} 1 & 1 & 1 & 1 & 1 & 1 \\ 0 & 1 & 0 & 1 & 1 & 0 \\ 1 & 1 & 1 & 1 & 1 & 1 \\ 0 & 0 & 0 & 1 & 1 & 0 \\ 0 & 0 & 0 & 1 & 1 & 0 \\ 0 & 1 & 0 & 1 & 1 & 1 \end{bmatrix}_{6 \times 6},$$

$$\mathbb{M}_{\bar{X}_3}^{\succeq B} = \begin{bmatrix} 1 & 0 & 0 & 1 & 0 & 1 & 0 \\ 1 & 1 & 0 & 1 & 0 & 1 & 1 \\ 1 & 1 & 1 & 1 & 1 & 1 & 1 \\ 0 & 0 & 0 & 1 & 0 & 1 & 0 \\ 1 & 1 & 0 & 1 & 1 & 1 & 1 \\ 0 & 0 & 0 & 1 & 0 & 1 & 0 \\ 1 & 0 & 0 & 1 & 0 & 1 & 1 \end{bmatrix}_{7 \times 7}.$$

*According to Corollary 1, the MCIM values are:*

$$MCIM_{(cl_1,cl_2)} = \frac{Sum(\mathbb{M}_{\overline{X}_1}^{\succeq B})}{25} = \frac{16}{25}, \tag{21}$$

$$MCIM_{(cl_1,cl_3)} = \frac{Sum(\mathbb{M}_{\overline{X}_2}^{\succeq B})}{36} = \frac{23}{36}, \tag{22}$$

$$MCIM_{(cl_2,cl_3)} = \frac{Sum(\mathbb{M}_{\overline{X}_3}^{\succeq B})}{49} = \frac{29}{49}. \tag{23}$$

*Then, by Corollary 2, the MCDS values are:*

$$MCDS_{(cl_1,cl_2)} = 1 - \frac{16}{25} = \frac{9}{25}, \tag{24}$$

$$MCDS_{(cl_1,cl_3)} = 1 - \frac{23}{36} = \frac{13}{36}, \tag{25}$$

$$MCDS_{(cl_2,cl_3)} = 1 - \frac{29}{49} = \frac{20}{49}. \tag{26}$$

*Finally, the MGDS is:* $MGDS_U(B, CL) = \frac{1}{3}\left(\frac{9}{25} + \frac{13}{36} + \frac{20}{49}\right) \approx 0.3764.$

**Corollary 4** (MGDS-based inner significance). *For any attribute $a \in B \subseteq C$, the inner significance of $a$ with respect to $B$ is defined by the change in MGDS after removing $a$:*

$$Msig_{inner}^{\succeq U}(a, B, d) = MGDS_U(B, CL) - MGDS_U(B \setminus \{a\}, CL). \tag{27}$$

*This measure is equivalent to the inner significance defined via the set-based GDS, yielding consistent results.*

**Corollary 5** (MGDS-based outer significance). *For any attribute $a \in C - B$, the outer significance of $a$ with respect to $B$ is defined by the increase in MGDS after adding $a$:*

$$Msig_{outer}^{\succeq U}(a, B, d) = MGDS_U(B \cup \{a\}, CL) - MGDS_U(B, CL). \tag{28}$$

*Similarly, this outer significance is consistent with the set-based GDS outer significance.*

Next, we introduce the definition of attribute reduction based on MGDS.

**Corollary 6** (Attribute reduct based on MGDS). *Let $S \succeq = (U, C \cup \{d\}, V, f)$ be an IODS, $\forall P \subseteq C$, the attribute subset $P$ is a reduct of $S \succeq$ if it satisfies*

*(1) $MGDS_U(P, CL) = MGDS_U(C, CL)$ and (2) $\forall a \in P, MGDS_U(P \setminus \{a\}, CL) \neq MGDS_U(P, CL)$.*

*The condition (1) ensures that the selected attribute subset maintains the same discriminative power as the entire attribute set; the condition (2) guarantees that each attribute in the subset is indispensable by removing any redundant attributes from it.*

## 5 Proposed method

This paper proposes an incremental mechanism for attribute reduction with object set variations. In an IODS, data dynamics manifest primarily in two forms: addition of new objects and deletion of existing ones. Both changes can impact the attribute reduction results, which are fundamentally dependent on the calculation of the MGDS. Computing MGDS from scratch upon every data update is computationally expensive, especially for large-scale datasets. To alleviate this, we propose two incremental update methods for efficiently recalculating MGDS.

### 5.1 The incremental method for obtaining attribute reduction when adding object set

We first present an incremental MGDS update for object addition in an IODS, followed by the corresponding incremental attribute reduction algorithm.

### 5.1.1 The incremental mechanism for adding object set in IODS

The core idea of the incremental update method is to update the expanded dominance matrix and the inter-class structure by incorporating the newly added objects without recomputing from scratch.

**Proposition 2** (Incremental update of expanded dominance matrix). *Let $S^\succeq = (U, A, V, f)$ be an IODS with universe $U = \{x_1, x_2, \ldots, x_n\}$. For any attribute subset $B \subseteq A$, assume the expanded dominance matrix on $U$ is $\mathbb{M}_U^{\succeq B} = [m_{(i,j)}^B]_{n \times n}$. Let $U^+ = \{x_{n+1}, x_{n+2}, \ldots, x_{n+n^+}\}$ be a set of newly added objects. Then, the updated expanded dominance matrix on $U \cup U^+$ with respect to $B$ is*

$$\mathbb{M}_{U \cup U^+}^{\succeq B} = [m_{(i,j)}^{+B}]_{(n+n^+) \times (n+n^+)}, \tag{29}$$

*where the entries are given by*

$$m_{(i,j)}^{+B} = \begin{cases} m_{(i,j)}^B, & \text{if} \quad 1 \leq i \wedge j \leq n, \\ 1, & \text{if} \quad x_j D_B^* x_i \wedge (i > n \vee j > n), \\ 0, & \text{otherwise.} \end{cases} \tag{30}$$

**Proposition 3** (Incremental update of inter-class). *Let the dominance inter-class set on $U$ be $Inter\_cl = \{(cl_i, cl_j) | 1 \leq i < j \leq T\}$. When adding new objects $U^+ = \{x_{n+1}, \ldots, x_{n+n^+}\}$, the updated inter-class set is*

$$Inter\_cl^+ = Inter\_cl \cup \{(cl_i, X_j) \mid cl_i \in CL, \ X_j \in U^+\} \tag{31}$$

*where each new object $X_j$ is assigned to its corresponding decision class based on its decision attribute value.*

### 5.1.2 Incremental attribute reduction algorithm for adding objects

Leveraging the above incremental updates, we propose the incremental attribute reduction algorithm MNAR-A as described in Algorithm 1.

The detailed description of the steps in Algorithm 1 and their time complexity are given as follows. Algorithm 1 can be roughly subdivided into six distinct steps as follows.

**Step 1. Initialization (lines 3–4)**: Updating the dataset and initializing variables takes $O(1)$ time.

**Step 2. Inter-class update (line 5)**: Updating the inter-class structure requires $O(|U'|)$ time.

**Step 3. Expanded dominance matrix update (line 6)**: Updating matrices costs $O(|C| \cdot |U'| \cdot |U^+|)$.

**Step 4. MGDS computation (line 7)**: Calculating MGDS values depends on matrix size, typically $O(|U^+|^2)$.

**Step 5. Attribute addition loop (lines 8–18)**: The greedy addition of attributes involves $O((|C \setminus P|) \cdot |U^+|^2)$ time.

**Step 6. Redundancy removal (lines 19–24)**: The iterative removal of redundant attributes has complexity $O(|P|^2 \cdot |U^+|^2)$.

Overall, the time complexity of MNAR-A is $O\big(|U'| + |C| \cdot |U'| \cdot |U^+| + (|C \setminus P|) \cdot |U^+|^2 + |P|^2 \cdot |U^+|^2\big)$.

### 5.2 The incremental method for obtaining attribute reduction when deleting object set

We present incremental MGDS updates principles for object deletions in an IODS. We also introduce MNAR-D, an incremental attribute-reduction algorithm for dynamic deletion, based on the principles.

---

**Algorithm 1** Incremental Attribute Reduction Algorithm for Adding Objects (MNAR-A).

---

1: **Input:**
(1) Original IODS $S^{\succeq} = (U, C \cup \{d\}, V, f)$ and added object set $U' = \{x_{n+1}, \ldots, x_{n+n^+}\}$.
(2) Original reduct $R_U$ on $U$, and expanded dominance matrices $\mathbb{M}_U^{\succeq C}$, $\mathbb{M}_U^{\succeq R_U}$.
(3) Original inter-class set $Inter\_cl$.
2: **Output:** Updated reduct $R_{U^+}$ on $U \cup U'$.
3: Initialize $P \leftarrow R_U$, $U^+ \leftarrow U \cup U'$.
4: Initialize $\mathbb{M}_{U^+}^{\succeq C} \leftarrow \mathbb{M}_U^{\succeq C}$, $\mathbb{M}_{U^+}^{\succeq P} \leftarrow \mathbb{M}_U^{\succeq R_U}$.
5: Update inter-class $Inter\_cl^+$ using Proposition 3.
6: Update expanded dominance matrices $\mathbb{M}_{U^+}^{\succeq C}$ and $\mathbb{M}_{U^+}^{\succeq P}$ using Proposition 2.
7: Compute $MGDS_{U^+}(P, CL)$ and $MGDS_{U^+}(C, CL)$ via Corollary 3.
8: **if** $MGDS_{U^+}(P, CL) = MGDS_{U^+}(C, CL)$ **then**
9:      Go to Line 24.
10: **else**
11:      **while** $MGDS_{U^+}(P, CL) \neq MGDS_{U^+}(C, CL)$ **do**
12:          **for** each $a_i \in C \setminus P$ **do**
13:              Compute $sig_{outer}^{\succeq U^+}(a_i, P, d)$ using Corollary 5.
14:          **end for**
15:          Select $a_0$ s.t. $a_0 \in \arg\max sig_{outer}^{\succeq U^+}(a_i, P, d)$.
16:          $P \leftarrow P \cup \{a_0\}$
17:      **end while**
18: **end if**
19: **for** each $a_i \in P$ **do**
20:      **if** $MGDS_{U^+}(P \setminus \{a_i\}, CL) = MGDS_{U^+}(C, CL)$ **then**
21:          $P \leftarrow P \setminus \{a_i\}$
22:      **end if**
23: **end for**
24: $R_{U^+} \leftarrow P$
25: **return** $R_{U^+}$

---

### 5.2.1 Incremental update of expanded dominance matrix and inter-class

Unlike the addition case, deleting objects from an IODS allows updating the expanded dominance matrix by removing corresponding rows and columns, avoiding a full recomputation. Similarly, the inter-class structure is updated by removing deleted objects from their respective decision classes.

**Proposition 4** (Update of expanded dominance matrix upon deletion). *Let $S^{\succeq} = (U, A, V, f)$ be an IODS with universe $U = \{x_1, x_2, \ldots, x_n\}$. For any attribute subset $B \subseteq A$, suppose the expanded dominance matrix on $U$ is $\mathbb{M}_U^{\succeq B} = [m_{(i,j)}^B]_{n \times n}$. Let the object subset $U^- = \{x_{y_1}, x_{y_2}, \ldots, x_{y_{n^-}}\}$, with indices ordered $1 \leq y_1 < y_2 < \cdots < y_{n^-} \leq n$, be deleted from $S^{\succeq}$. Then the updated expanded dominance matrix on the reduced universe $U - U^-$ with respect to $B$ is*

$$\mathbb{M}_{U \setminus U^-}^{\succeq B} = [m_{(i,j)}^{-B}]_{(n-n^-) \times (n-n^-)}, \tag{32}$$

*where the entries are given by*

$$m_{(i,j)}^{-B} = \begin{cases} m_{(i+k_s-1, \, j+k_l-1)}^B, & \text{case}\,(1) \\ m_{(i+k_s-1, \, j+n^-)}^B, & \text{case}\,(2) \\ m_{(i+n^-, \, j+k_l-1)}^B, & \text{case}\,(3) \\ m_{(i+n^-, \, j+n^-)}^B, & \text{case}\,(4) \end{cases}$$

$$
\begin{aligned}
\text{case}(1): \quad & y_{k_s-1} - k_s + 2 \le i, j < y_{k_s} - k_s + 1 \\
\text{case}(2): \quad & y_{k_s-1} - k_s + 2 \le i < y_{k_s} - k_s + 1, \\
& y_{n^-} - n^- + 1 \le j \le n - n^-, \\
\text{case}(3): \quad & y_{n^-} - n^- + 1 \le i \le n - n^-, \\
& y_{k_l-1} - k_l + 2 \le j < y_{k_l} - k_l + 1, \\
\text{case}(4): \quad & y_{n^-} - n^- + 1 \le i, j \le n - n^-,
\end{aligned}
$$

*for $1 \le k_s, k_l \le n^-$, and with the convention $y_0 = 0$.*

**Proposition 5** (Update of inter-class upon deletion)**.** *Let the original dominance inter-class set be $Inter\_cl = \{(cl_i, cl_j) \mid 1 \le i < j \le T\}$. Upon deleting objects $U^- \subseteq U$, the updated inter-class set is*

$$
Inter\_cl^- = \{(cl_i^-, cl_j^-) \mid 1 \le i < j \le T\}, \tag{33}
$$

*where each updated decision class is $cl_i^- = cl_i \setminus \{x \in U^- \mid x \in cl_i\}$.*

### 5.2.2 Incremental attribute reduction algorithm for deleting objects

Based on the above update rules, we design an incremental algorithm, MNAR-D, to efficiently update attribute reducts after object deletions.

---

**Algorithm 2** Incremental Attribute Reduction Algorithm for Deleting Objects (MNAR-D).

---

1: **Input:**
  (1) Original IODS $S^{\succeq} = (U, C \cup \{d\}, V, f)$, deleted object set $U^- = \{x_{y_1}, x_{y_2}, \ldots, x_{y_{n^-}}\}$.
  (2) Original reduct $R_U$ on $U$, and expanded dominance matrices $\mathbb{M}_U^{\succeq C}$, $\mathbb{M}_U^{\succeq R_U}$.
  (3) Original inter-class set $Inter\_cl$.
2: **Output:** Updated reduct $R^-$ on $U \setminus U^-$.
3: Initialize $P \leftarrow R_U$, $\mathbb{M}_{U \setminus U^-}^{\succeq C} \leftarrow \mathbb{M}_U^{\succeq C}$, $\mathbb{M}_{U \setminus U^-}^{\succeq P} \leftarrow \mathbb{M}_U^{\succeq R_U}$, and $Inter\_cl^- \leftarrow Inter\_cl$.
4: Update inter-class $Inter\_cl^-$ using Proposition 5.
5: Update expanded dominance matrices $\mathbb{M}_{U-U^-}^{\succeq C}$ and $\mathbb{M}_{U \setminus U^-}^{\succeq P}$ using Proposition 4.
6: Compute $MGDS_{U \setminus U^-}(P, CL)$ and $MGDS_{U \setminus U^-}(C, CL)$ via Corollary 3.
7: **if** $MGDS_{U \setminus U^-}(P, CL) = MGDS_{U \setminus U^-}(C, CL)$ **then**
8:    Go to Step 18.
9: **else**
10:    **while** $MGDS_{U \setminus U^-}(P, CL) \ne MGDS_{U \setminus U^-}(C, CL)$ **do**
11:       **for** each $a_i \in C \setminus P$ **do**
12:          Compute $sig_{outer}^{\succeq U \setminus U^-}(a_i, P, d)$ using Corollary 5.
13:       **end for**
14:       Select $a_0$ s.t. $a_0 \in \arg\max sig_{outer}^{\succeq U \setminus U^-}(a_i, P, d)$.
15:       $P \leftarrow P \cup \{a_0\}$
16:    **end while**
17: **end if**
18: **for** each $a_i \in P$ **do**
19:    **if** $MGDS_{U \setminus U^-}(P \setminus \{a_i\}, CL) = MGDS_{U \setminus U^-}(C, CL)$ **then**
20:       $P \leftarrow P \setminus \{a_i\}$
21:    **end if**
22: **end for**
23: $R^- \leftarrow P$
24: **return** $R^-$

---

The detailed description of the steps in Algorithm 2 and their time complexity are given as follows. Algorithm 2 consists of the following six distinct steps.

**Step 1. Initialization (line 3):** $O(1)$.

**Step 2. Inter-class update (line 4):** $O(|U^-|)$.

**Step 3. Expanded dominance matrix update (line 5):** $O(|U| \times |U^-|)$.

**Step 4. MGDS computation (line 6):** depending on reduced matrix size, roughly $O(|U \setminus U^-|^2)$.

**Step 5. Attribute addition loop (lines 7–17):** $O((|C \setminus P|) \times |U \setminus U^-|^2)$.

**Step 6. Redundancy removal (lines 18–23):** $O(|P|^2 \times |U \setminus U^-|^2)$.

Overall complexity is $O\big(|U^-| + |U| \cdot |U^-| + (|C \setminus P|)|U \setminus U^-|^2 + |P|^2|U \setminus U^-|^2\big)$, which is typically much lower than recomputing from scratch.

## 6 Experimental evaluation

We assess the proposed methods from the following perspectives.

- Comparison of classification accuracy with existing feature selection algorithms under the same sample conditions.

- Comparison of the time consumed by the algorithm when the number of samples increases or decreases.

The effectiveness of the proposed algorithm is evaluated through extensive experiments across nine different datasets from the UCI Machine Learning Repository Kelly et al. (2023) as described in Table 2. To meet the needs of the IODS algorithm, the data preprocessing includes the following steps. First, we identify and remove completely redundant duplicate samples from the dataset to avoid overfitting. Second, to simulate the issue of missing data in real-world scenarios, 30% of the conditional attributes are randomly selected for each dataset while ensuring the structural integrity of the dataset, and 10% of the sample values within these selected attributes are randomly marked as missing values ("*"). All algorithms in this paper are implemented in Python and are executed on a computer equipped with a 3.0 GHz Intel® Core™ i9-13900 CPU, 128 GB of RAM, and a 64-bit Windows 11 operating system.

Table 2: Details of the nine UCI datasets. "Objects" refers to the number of instances, "attributes" denotes the number of conditional attributes, and "classes" indicates the number of decision categories.

| Dataset | Objects | Attributes | Classes |
|---------|---------|------------|---------|
| Iris | 150 | 4 | 3 |
| HD | 303 | 13 | 5 |
| BCWO | 699 | 9 | 2 |
| NPHA | 714 | 14 | 3 |
| Statlog | 946 | 18 | 4 |
| Car | 1728 | 6 | 4 |
| Rice | 3810 | 7 | 2 |
| Predict | 4424 | 36 | 3 |
| Wine | 4898 | 11 | 7 |

### 6.1 Performance evaluation of MNAR-A

We evaluate the performance of algorithm MNAR-A from the perspective of effectiveness and efficiency. In addition to comparing the incremental algorithm MNAR-A with the non-incremental algorithm MNAR, we also selected four existing attribute reduction algorithms, IAR Sang et al. (2021), IAR-A Sang et al.

(2021), IDDC Ullah et al. (2024), and a method by Du et al. Du & Hu (2016), for comparison. IAR is a non-incremental attribute reduction algorithm based on dominance conditional entropy. IAR-A is an incremental attribute reduction algorithm based on dominance condition entropy when objects are added. IDDC is an attribute reduction algorithm based on incremental dominance dependency, while the method by Du et al. is a heuristic attribute reduction algorithm based on characteristic dominance relations.

### 6.1.1 Effectiveness

To simulate dynamic datasets, the randomly selected 50% objects of each dataset were used as the original object set, and the remaining 50% as newly added objects. Attribute reductions were performed by each algorithm on the updated dataset. The classification performance of the reduced datasets was evaluated using SVM, KNN, and Random Forest (RF) classifiers with ten-fold cross-validation. The results are reported in Table 3–5.

MNAR-A performed comparable or outperforming the competing algorithms on the majority of datasets and classifiers, both in per-dataset scores and overall averages. These findings verify that MNAR-A delivers high-quality attribute reducts after object insertions.

Table 3: Classification accuracy comparison based on KNN (mean ± std).

| Dataset | IAR | IAR-A | IDDC | Du et al. | MNAR | **MNAR-A** |
|---|---|---|---|---|---|---|
| Iris | 0.9463±0.0470 | 0.9533±0.0521 | 0.9533±0.0340 | 0.8800±0.1483 | 0.9533±0.0340 | **0.9667±0.0211** |
| HD | 0.4972±0.0643 | 0.4999±0.0770 | 0.5116±0.0463 | 0.5112±0.0448 | 0.5095±0.0674 | **0.5212±0.0539** |
| BCWO | 0.9571±0.0322 | **0.9657±0.0345** | 0.9642±0.0221 | 0.8381±0.0278 | 0.9600±0.0273 | 0.9484±0.0379 |
| NPHA | 0.5370±0.0493 | 0.5665±0.0307 | 0.5482±0.0252 | 0.5314±0.0413 | 0.5777±0.0380 | **0.6402±0.0486** |
| Statlog | 0.6321±0.0327 | 0.6525±0.0375 | 0.6253±0.0373 | 0.5901±0.0755 | **0.6443±0.0442** | 0.6101±0.0439 |
| Car | 0.8056±0.0547 | 0.8397±0.0512 | 0.8131±0.0526 | 0.7186±0.0466 | 0.8056±0.0547 | **0.8588±0.0023** |
| Rice | 0.8714±0.0283 | 0.8816±0.0367 | 0.8706±0.0318 | 0.8828±0.1161 | 0.9160±0.0227 | **0.9165±0.0160** |
| Predict | 0.5873±0.0124 | 0.5992±0.0079 | 0.5047±0.0749 | 0.5135±0.0128 | 0.6155±0.0129 | **0.6581±0.0296** |
| Wine | 0.6014±0.0173 | 0.6092±0.0221 | 0.4967±0.0111 | 0.4501±0.1232 | 0.6177±0.0281 | **0.6087±0.0385** |
| Average | 0.7150±0.0376 | 0.7297±0.0389 | 0.6986±0.0373 | 0.6573±0.0707 | 0.7333±0.0366 | **0.7476±0.0324** |

Table 4: Classification accuracy comparison based on SVM (mean ± std).

| Dataset | IAR | IAR-A | IDDC | Du et al. | MNAR | **MNAR-A** |
|---|---|---|---|---|---|---|
| Iris | 0.9532±0.0225 | 0.9600±0.0327 | 0.9467±0.0499 | 0.8667±0.1633 | 0.9600±0.0327 | **0.9762±0.0127** |
| HD | 0.5078±0.0127 | 0.5397±0.0062 | 0.5412±0.0108 | 0.5412±0.0108 | 0.5412±0.0108 | **0.5563±0.0135** |
| BCWO | 0.9643±0.0271 | 0.9628±0.0294 | 0.9714±0.0300 | 0.8924±0.0124 | **0.9671±0.0200** | 0.9627±0.0214 |
| NPHA | 0.6281±0.0352 | 0.6210±0.0054 | 0.6211±0.0172 | 0.6211±0.0212 | 0.6412±0.0172 | **0.6813±0.0325** |
| Statlog | 0.4872±0.0218 | 0.4965±0.0338 | 0.4870±0.0299 | 0.4621±0.0693 | **0.5000±0.0297** | 0.4851±0.0255 |
| Car | 0.8739±0.0487 | 0.8739±0.0487 | 0.8895±0.0435 | 0.6081±0.1837 | 0.8739±0.0487 | **0.9136±0.0137** |
| Rice | 0.8765±0.0332 | 0.8808±0.0329 | 0.8801±0.0393 | **0.9577±0.0411** | 0.8803±0.0327 | 0.9123±0.0146 |
| Predict | 0.4993±0.0006 | 0.4993±0.0006 | 0.4687±0.0015 | 0.4993±0.0010 | 0.4993±0.0010 | **0.5335±0.0006** |
| Wine | 0.5417±0.0076 | 0.5434±0.0085 | 0.5420±0.0180 | 0.5488±0.0015 | 0.5420±0.0180 | **0.5463±0.0103** |
| Average | 0.7036±0.0233 | 0.7086±0.0220 | 0.7053±0.0267 | 0.6664±0.0560 | 0.7117±0.0234 | **0.7297±0.0161** |

Table 5: Classification accuracy comparison based on RF (mean ± std).

| Dataset | IAR | IAR-A | IDDC | Du et al. | MNAR | **MNAR-A** |
|---|---|---|---|---|---|---|
| Iris | 0.9667±0.0211 | 0.9533±0.0600 | 0.9467±0.0581 | 0.9333±0.0843 | 0.9533±0.0267 | **0.9733±0.0211** |
| HD | 0.5846±0.0276 | 0.5849±0.0343 | 0.5904±0.0579 | 0.6073±0.0473 | 0.5542±0.0447 | **0.6275±0.0376** |
| BCWO | 0.9571±0.0342 | 0.9557±0.0314 | 0.9700±0.0251 | 0.9586±0.0257 | 0.9628±0.0199 | **0.9813±0.0149** |
| NPHA | 0.5482±0.0546 | 0.5342±0.0278 | 0.5495±0.0382 | 0.6266±0.0163 | 0.6182±0.0113 | **0.7064±0.0175** |
| Statlog | 0.7372±0.0158 | 0.7447±0.0165 | 0.7494±0.0278 | 0.7360±0.0495 | 0.7228±0.0515 | **0.7938±0.0125** |
| Car | 0.7066±0.0134 | 0.8200±0.0524 | **0.8698±0.0618** | 0.6776±0.0132 | 0.7066±0.0134 | 0.8588±0.0113 |
| Rice | 0.9131±0.0228 | 0.9207±0.0181 | 0.9194±0.0217 | 0.9008±0.0236 | 0.9249±0.0193 | **0.9735±0.0176** |
| Predict | 0.7107±0.0075 | 0.7731±0.0061 | 0.7749±0.0081 | 0.7208±0.0098 | 0.7380±0.0078 | **0.7842±0.0057** |
| Wine | 0.6136±0.0164 | 0.6206±0.0237 | 0.6286±0.0338 | 0.5904±0.0273 | 0.6163±0.0405 | **0.6906±0.0059** |
| Average | 0.7486±0.0220 | 0.7675±0.0300 | 0.7776±0.0369 | 0.7502±0.0330 | 0.7552±0.0261 | **0.8210±0.0160** |

### 6.1.2 Sensitivity Experiments on different attribute and object Missing Rates

In order to evaluate how attribute missing rates and sample value missing rates affect the results, we performed experiments using different missing rates on three datasets: BCWO (small attribute count), HD (medium attribute count), and predict (large attribute count).

With the sample missing rate fixed at 10%, we adjusted the attribute missing rate from the original 30% to 40% and 50% to observe the impact. Conversely, while keeping the attribute missing rate constant at 30%, we increased the sample missing rate from the original 10% to 20% and 30%.The results are reported in Table 6–17.

When the sample missing rate and attribute missing rate vary, MNAR-A exhibits only a slight drop in accuracy, and consistently outperforms other methods in terms of classification accuracy under various combinations of sample and attribute missing rates. These results further demonstrate the effectiveness and robustness of our proposed approach.

Table 6: Classification Accuracy Comparison based on KNN under 10% Sample value Missing Rate and 40% Attribute Missing Rate (mean ± std).

| Dataset | IAR | IAR-A | IDDC | Du et al. | MNAR | **MNAR-A** |
|---|---|---|---|---|---|---|
| BCWO | 0.9185±0.0330 | 0.9272±0.0350 | 0.9256±0.0230 | 0.7826±0.0300 | 0.9210±0.0300 | **0.9335±0.0312** |
| HD | 0.4724±0.0660 | 0.4751±0.0790 | 0.4868±0.0480 | 0.4862±0.0470 | 0.4847±0.0690 | **0.4965±0.0560** |
| Predict | 0.5436±0.0140 | 0.5548±0.0090 | 0.4602±0.0770 | 0.4680±0.0150 | 0.5728±0.0150 | **0.6252±0.0310** |
| Average | 0.6448±0.0377 | 0.6524±0.0410 | 0.6242±0.0493 | 0.5789±0.0307 | 0.6595±0.0380 | **0.6851±0.0394** |

Table 7: Classification Accuracy Comparison based on KNN under 10% Sample value Missing Rate and 50% Attribute Missing Rate (mean ± std).

| Dataset | IAR | IAR-A | IDDC | Du et al. | MNAR | **MNAR-A** |
|---|---|---|---|---|---|---|
| BCWO | 0.8901±0.0340 | 0.8987±0.0360 | 0.8971±0.0240 | 0.7461±0.0320 | 0.8925±0.0320 | **0.9018±0.0335** |
| HD | 0.4477±0.0680 | 0.4504±0.0810 | 0.4621±0.0500 | 0.4615±0.0490 | 0.4599±0.0710 | **0.4738±0.0580** |
| Predict | 0.5109±0.0160 | 0.5205±0.0110 | 0.4357±0.0800 | 0.4325±0.0170 | 0.5401±0.0170 | **0.5995±0.0330** |
| Average | 0.6162±0.0393 | 0.6232±0.0427 | 0.5983±0.0513 | 0.5467±0.0327 | 0.6308±0.0400 | **0.6584±0.0415** |

Table 8: Classification Accuracy Comparison based on KNN under 20% Sample value Missing Rate and 30% Attribute Missing Rate (mean ± std).

| Dataset | IAR | IAR-A | IDDC | Du et al. | MNAR | **MNAR-A** |
|---|---|---|---|---|---|---|
| BCWO | 0.9126±0.0330 | 0.9218±0.0350 | 0.9203±0.0230 | 0.7915±0.0300 | 0.9150±0.0300 | **0.9148±0.0310** |
| HD | 0.4589±0.0660 | 0.4615±0.0790 | 0.4751±0.0480 | 0.4745±0.0470 | 0.4708±0.0690 | **0.4897±0.0560** |
| Predict | 0.5498±0.0140 | 0.5605±0.0090 | 0.4693±0.0770 | 0.4762±0.0150 | 0.5788±0.0150 | **0.6262±0.0310** |
| Average | 0.6404±0.0377 | 0.6479±0.0410 | 0.6216±0.0493 | 0.5807±0.0307 | 0.6549±0.0380 | **0.6769±0.0393** |

Table 9: Classification Accuracy Comparison based on KNN under 30% Sample value Missing Rate and 30% Attribute Missing Rate(mean ± std).

| Dataset | IAR | IAR-A | IDDC | Du et al. | MNAR | **MNAR-A** |
|---|---|---|---|---|---|---|
| BCWO | 0.9051±0.0340 | 0.9143±0.0360 | 0.9128±0.0240 | 0.7842±0.0320 | 0.9075±0.0320 | **0.9095±0.0320** |
| HD | 0.4514±0.0680 | 0.4540±0.0810 | 0.4684±0.0500 | 0.4678±0.0490 | 0.4633±0.0710 | **0.4844±0.0580** |
| Predict | 0.5431±0.0160 | 0.5528±0.0110 | 0.4626±0.0800 | 0.4687±0.0170 | 0.5713±0.0170 | **0.6205±0.0330** |
| Average | 0.6332±0.0393 | 0.6404±0.0427 | 0.6146±0.0513 | 0.5736±0.0327 | 0.6474±0.0400 | **0.6715±0.0410** |

Table 10: Classification Accuracy Comparison based on SVM under 10% Sample value Missing Rate and 40% Attribute Missing Rate (mean ± std).

| Dataset | IAR | IAR-A | IDDC | Du et al. | MNAR | **MNAR-A** |
|---|---|---|---|---|---|---|
| BCWO | 0.8915±0.0290 | 0.8892±0.0310 | 0.8984±0.0320 | 0.8189±0.0150 | 0.8941±0.0230 | **0.9127±0.0240** |
| HD | 0.4568±0.0104 | 0.4887±0.0244 | 0.4902±0.0112 | 0.4892±0.0001 | 0.4902±0.0112 | **0.5093±0.0114** |
| Predict | 0.4393±0.0223 | 0.4393±0.0258 | 0.4037±0.0233 | 0.4343±0.0000 | 0.4393±0.0222 | **0.4835±0.0321** |
| Average | 0.5959±0.0206 | 0.6057±0.0271 | 0.5974±0.0222 | 0.5808±0.0050 | 0.6079±0.0188 | **0.6352±0.0225** |

Table 11: Classification Accuracy Comparison based on SVM under 10% Sample value Missing Rate and 50% Attribute Missing Rate (mean ± std).

| Dataset | IAR | IAR-A | IDDC | Du et al. | MNAR | **MNAR-A** |
|---|---|---|---|---|---|---|
| BCWO | 0.8635±0.0300 | 0.8612±0.0320 | 0.8704±0.0330 | 0.7839±0.0170 | 0.8661±0.0250 | **0.8877±0.0260** |
| HD | 0.4268±0.0156 | 0.4587±0.0328 | 0.4602±0.0113 | 0.4592±0.0113 | 0.4602±0.0113 | **0.4843±0.0115** |
| Predict | 0.4093±0.0322 | 0.4593±0.0423 | 0.3937±0.0423 | 0.4043±0.0323 | 0.4093±0.0103 | **0.4585±0.0356** |
| Average | 0.5665±0.0259 | 0.5931±0.0357 | 0.5748±0.0289 | 0.5491±0.0202 | 0.5785±0.0155 | **0.6102±0.0244** |

Table 12: Classification Accuracy Comparison based on SVM under 20% Sample value Missing Rate and 30% Attribute Missing Rate (mean ± std).

| Dataset | IAR | IAR-A | IDDC | Du et al. | MNAR | **MNAR-A** |
|---|---|---|---|---|---|---|
| BCWO | 0.9195±0.0280 | 0.9172±0.0300 | 0.9264±0.0310 | 0.8469±0.0140 | 0.9221±0.0220 | **0.9377±0.0230** |
| HD | 0.4768±0.0113 | 0.5187±0.0620 | 0.5202±0.0411 | 0.5192±0.0311 | 0.5202±0.0111 | **0.5333±0.0113** |
| Predict | 0.4643±0.0231 | 0.4643±0.0311 | 0.4337±0.0522 | 0.4643±0.0211 | 0.4643±0.0211 | **0.5085±0.0211** |
| Average | 0.6202±0.0208 | 0.6334±0.0410 | 0.6268±0.0414 | 0.6101±0.0221 | 0.6355±0.0181 | **0.6598±0.0185** |

Table 13: Classification Accuracy Comparison based on SVM under 30% Sample value Missing Rate and 30% Attribute Missing Rate (mean ± std).

| Dataset | IAR | IAR-A | IDDC | Du et al. | MNAR | **MNAR-A** |
|---|---|---|---|---|---|---|
| BCWO | 0.9135±0.0290 | 0.9112±0.0310 | 0.9204±0.0320 | 0.8409±0.0150 | 0.9161±0.0230 | **0.9347±0.0240** |
| HD | 0.4708±0.0214 | 0.5127±0.0217 | 0.5142±0.0112 | 0.5132±0.0112 | 0.5142±0.0112 | **0.5303±0.0114** |
| Predict | 0.4583±0.0322 | 0.4583±0.0322 | 0.4277±0.0321 | 0.4583±0.0223 | 0.4583±0.0220 | **0.5055±0.0126** |
| Average | 0.6142±0.0275 | 0.6274±0.0283 | 0.6208±0.0251 | 0.6041±0.0162 | 0.6295±0.0187 | **0.6568±0.0160** |

Table 14: Classification Accuracy Comparison based on RF under 10% Sample value Missing Rate and 40% Attribute Missing Rate (mean ± std).

| Dataset | IAR | IAR-A | IDDC | Du et al. | MNAR | **MNAR-A** |
|---|---|---|---|---|---|---|
| BCWO | 0.8971±0.0360 | 0.8957±0.0330 | 0.9100±0.0270 | 0.9086±0.0270 | 0.9028±0.0210 | **0.9333±0.0170** |
| HD | 0.5446±0.0290 | 0.5449±0.0360 | 0.5504±0.0600 | 0.5773±0.0490 | 0.5142±0.0460 | **0.5795±0.0390** |
| Predict | 0.6307±0.0090 | 0.6931±0.0080 | 0.6949±0.0100 | 0.6508±0.0110 | 0.6580±0.0090 | **0.7282±0.0070** |
| Average | 0.6908±0.0247 | 0.7112±0.0257 | 0.7184±0.0323 | 0.7122±0.0290 | 0.6917±0.0253 | **0.7470±0.0210** |

Table 15: Classification Accuracy Comparison based on RF under 10% Sample value Missing Rate and 50% Attribute Missing Rate (mean ± std).

| Dataset | IAR | IAR-A | IDDC | Du et al. | MNAR | **MNAR-A** |
|---|---|---|---|---|---|---|
| BCWO | 0.8721±0.0372 | 0.8707±0.0340 | 0.8850±0.0280 | 0.8836±0.0280 | 0.8778±0.0220 | **0.9133±0.0180** |
| HD | 0.5246±0.0366 | 0.5249±0.0370 | 0.5304±0.0620 | 0.5573±0.0500 | 0.4942±0.0480 | **0.5595±0.0410** |
| Predict | 0.6007±0.0123 | 0.6631±0.0090 | 0.6649±0.0110 | 0.6208±0.0120 | 0.6280±0.0100 | **0.7032±0.0080** |
| Average | 0.6658±0.0287 | 0.6862±0.0267 | 0.6934±0.0337 | 0.6872±0.0300 | 0.6667±0.0267 | **0.7253±0.0223** |

Table 16: Classification Accuracy Comparison based on RF under 20% Sample value Missing Rate and 30% Attribute Missing Rate (mean ± std).

| Dataset | IAR | IAR-A | IDDC | Du et al. | MNAR | **MNAR-A** |
|---|---|---|---|---|---|---|
| BCWO | 0.9171±0.0350 | 0.9157±0.0320 | 0.9300±0.0260 | 0.9286±0.0260 | 0.9228±0.0200 | **0.9503±0.0160** |
| HD | 0.5646±0.0280 | 0.5649±0.0350 | 0.5704±0.0590 | 0.5923±0.0480 | 0.5342±0.0450 | **0.6075±0.0380** |
| Predict | 0.6557±0.0080 | 0.7181±0.0070 | 0.7199±0.0090 | 0.6758±0.0100 | 0.6730±0.0080 | **0.7512±0.0060** |
| Average | 0.7125±0.0237 | 0.7329±0.0247 | 0.7401±0.0313 | 0.7322±0.0280 | 0.7100±0.0243 | **0.7697±0.0200** |

Table 17: Classification Accuracy Comparison based on RF under 30% Sample value Missing Rate and 30% Attribute Missing Rate (mean ± std).

| Dataset | IAR | IAR-A | IDDC | Du et al. | MNAR | **MNAR-A** |
|---|---|---|---|---|---|---|
| BCWO | 0.9101±0.0360 | 0.9087±0.0330 | 0.9230±0.0270 | 0.9216±0.0270 | 0.9158±0.0210 | **0.9463±0.0170** |
| HD | 0.5576±0.0290 | 0.5579±0.0360 | 0.5634±0.0600 | 0.5853±0.0490 | 0.5272±0.0460 | **0.6015±0.0390** |
| Predict | 0.6487±0.0090 | 0.7111±0.0080 | 0.7129±0.0100 | 0.6688±0.0110 | 0.6660±0.0090 | **0.7452±0.0070** |
| Average | 0.7055±0.0247 | 0.7259±0.0257 | 0.7331±0.0323 | 0.7252±0.0290 | 0.7030±0.0253 | **0.7643±0.0210** |

### 6.1.3 Efficiency

For each dataset, the initial object set comprised the randomly 50% of objects. The remaining objects were then inserted in five stages: 20%, 40%, 60%, 80%, and 100%, runtime of MNAR-A and the other five methods was recorded after each insertion. Figure 1- 8 plots the resulting runtime curves.

MNAR-A consistently outperformed the others in running time due to leveraging prior computations. Attributed to its simpler dominance matrix calculations compared to the entropy-based computations in IAR-A. For Non-incremental methods, particularly IDDC, exhibited substantially longer running times due to retraining from scratch and complex dependency calculations.

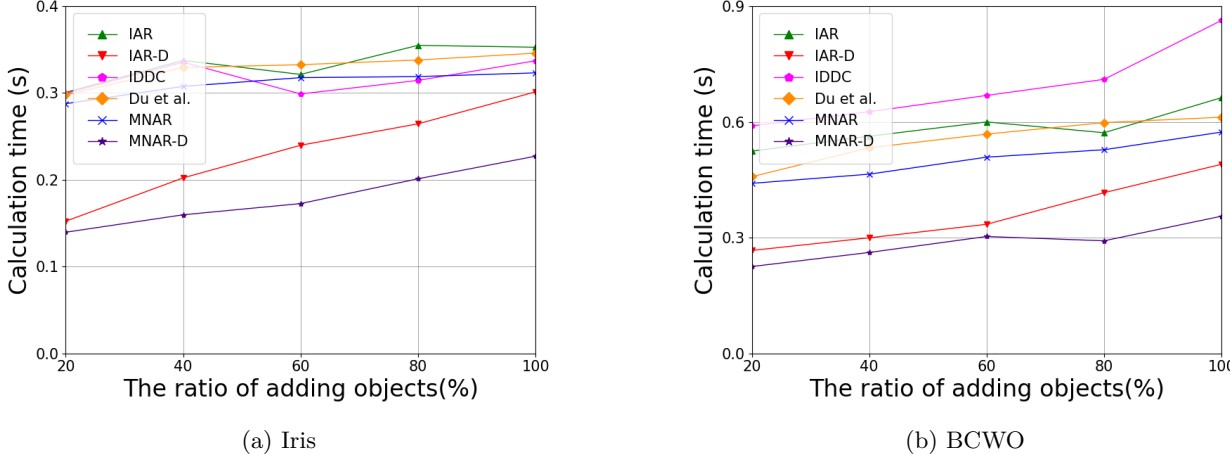

(a) Iris

(b) BCWO

Figure 1: Computation time on Iris and BCWO datasets at varying adding object ratios.

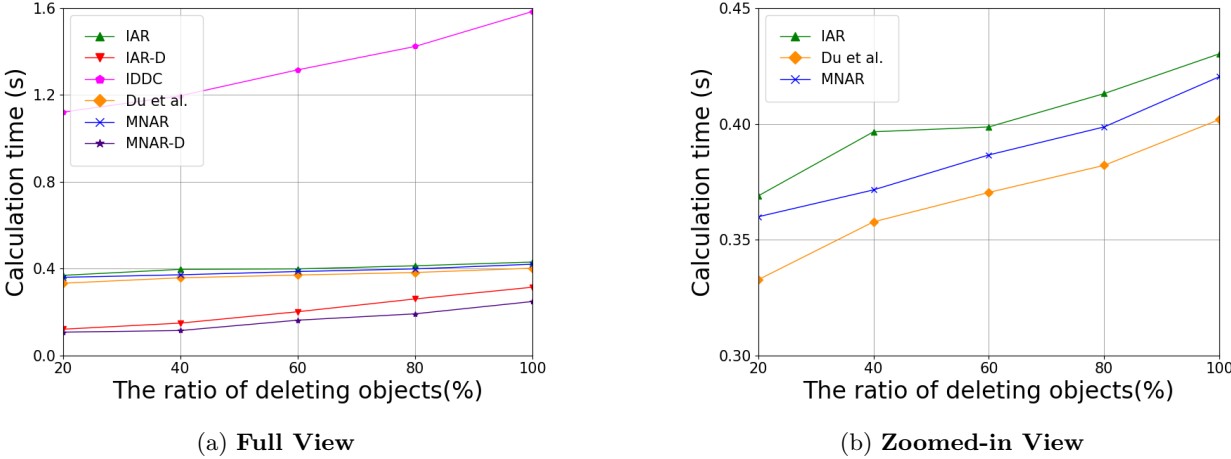

(a) **Full View**

(b) **Zoomed-in View**

Figure 2: Computation time on the **HD dataset** at varying added object ratios. (**Right**: Zoomed-in view of the $0.3 - 0.45$ s range to distinguish lower-running-time methods.)

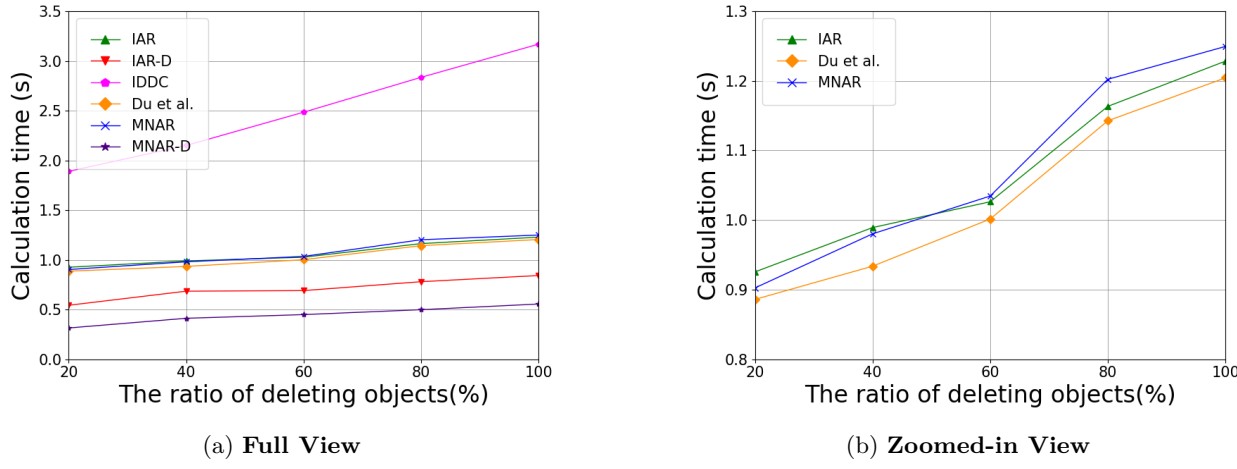

(a) **Full View**

(b) **Zoomed-in View**

Figure 3: Computation time on the **NPHA dataset** at varying added object ratios. (**Right**: Zoomed-in view of the $0.8 - 1.3$ s range to distinguish lower-running-time methods.)

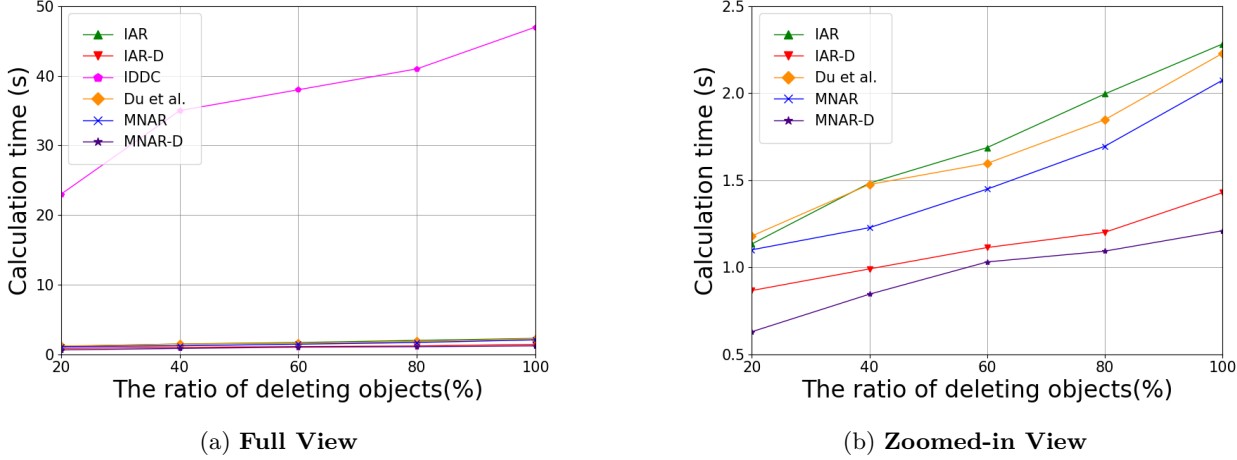

(a) **Full View**

(b) **Zoomed-in View**

Figure 4: Computation time on the **Stalog dataset** at varying added object ratios. (**Right**: Zoomed-in view of the $0.5 - 2.5$ s range to distinguish lower-running-time methods.)

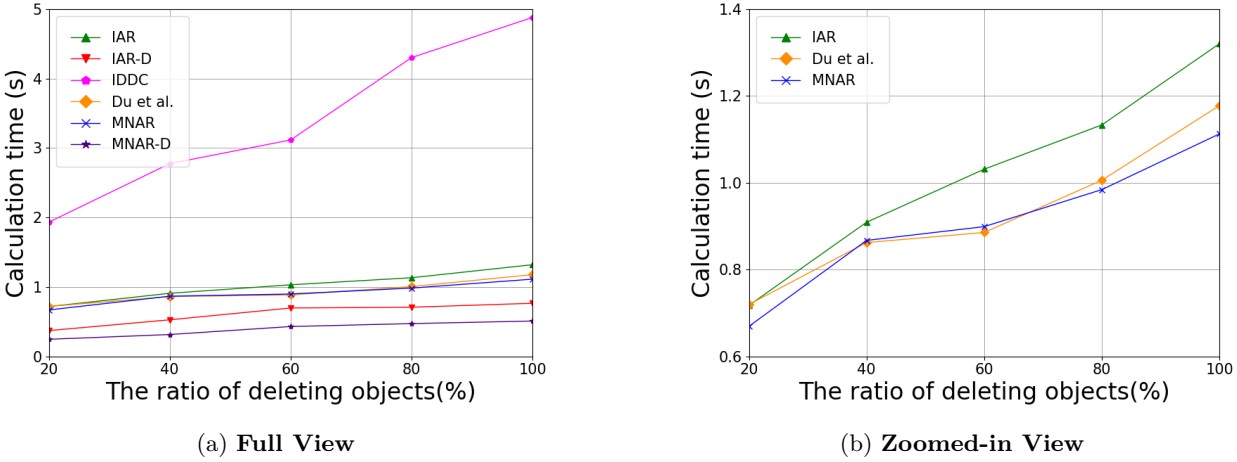

(a) **Full View**

(b) **Zoomed-in View**

Figure 5: Computation time on the **Car dataset** at varying added object ratios. (**Right**: Zoomed-in view of the $0.6 - 1.4$ s range to distinguish lower-running-time methods.)

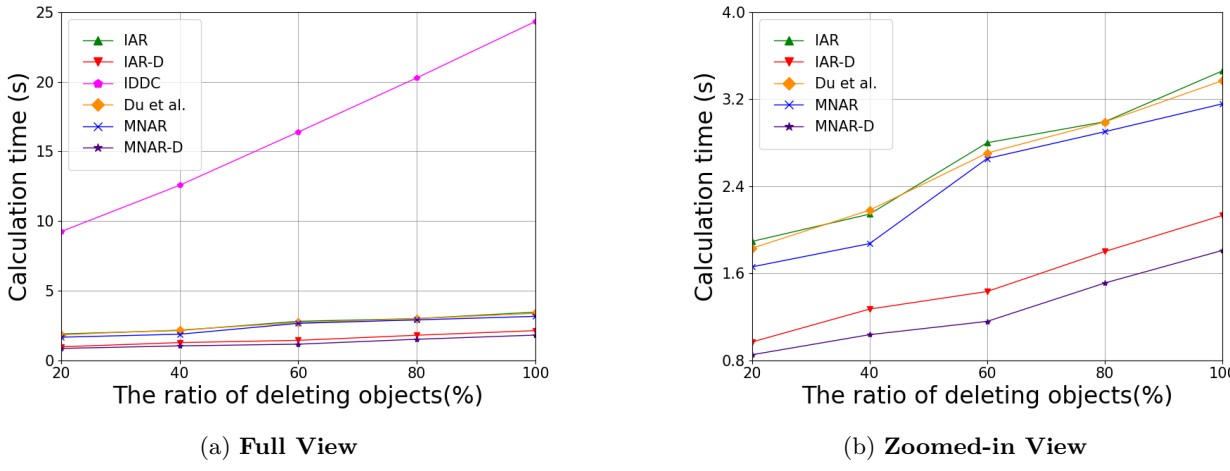

(a) **Full View**  (b) **Zoomed-in View**

Figure 6: Computation time on the **Rice dataset** at varying added object ratios. (**Right**: Zoomed-in view of the $0.8 - 4.0$ s range to distinguish lower-running-time methods.)

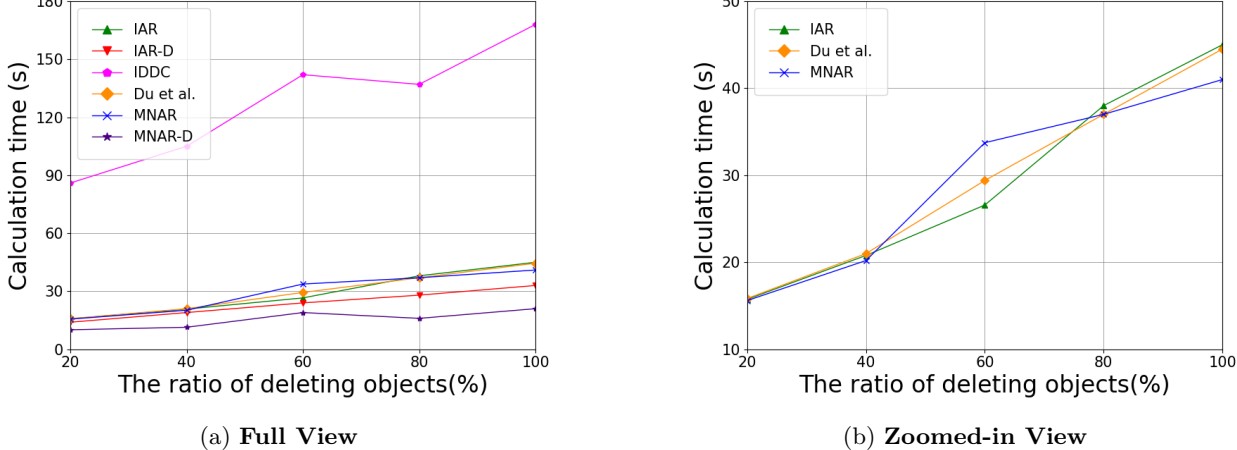

(a) **Full View**  (b) **Zoomed-in View**

Figure 7: Computation time on the **Predict dataset** at varying added object ratios. (**Right**: Zoomed-in view of the $10 - 50$ s range to distinguish lower-running-time methods.)

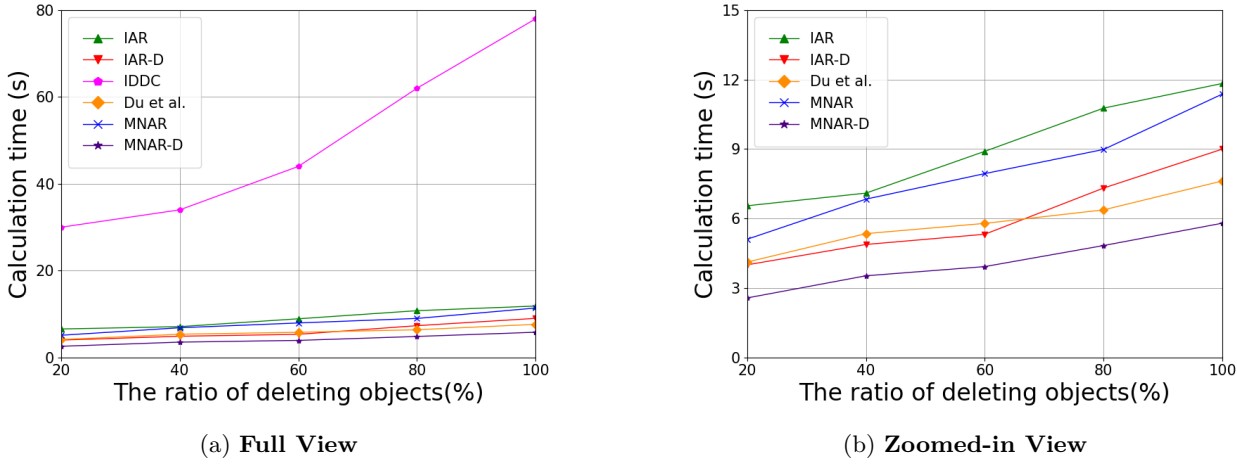

(a) **Full View**  (b) **Zoomed-in View**

Figure 8: Computation time on the **Wine dataset** at varying added object ratios. (**Right**: Zoomed-in view of the $0 - 15$ s range to distinguish lower-running-time methods.)

Figure 9- 11 further illustrates the speedup ratios of MNAR-A over the competing methods across different object addition stages on various dataset. The speedup is defined as the ratio between the runtime of each baseline method and that of MNAR-A under the same added-object ratio.

As shown in the figure, MNAR-A consistently achieves speedup values greater than 100% on all datasets and at all insertion stages, indicating a stable and persistent efficiency advantage throughout the incremental process. Although the absolute magnitude of the speedup varies across datasets, the overall trends remain consistent, without noticeable degradation as more objects are added.

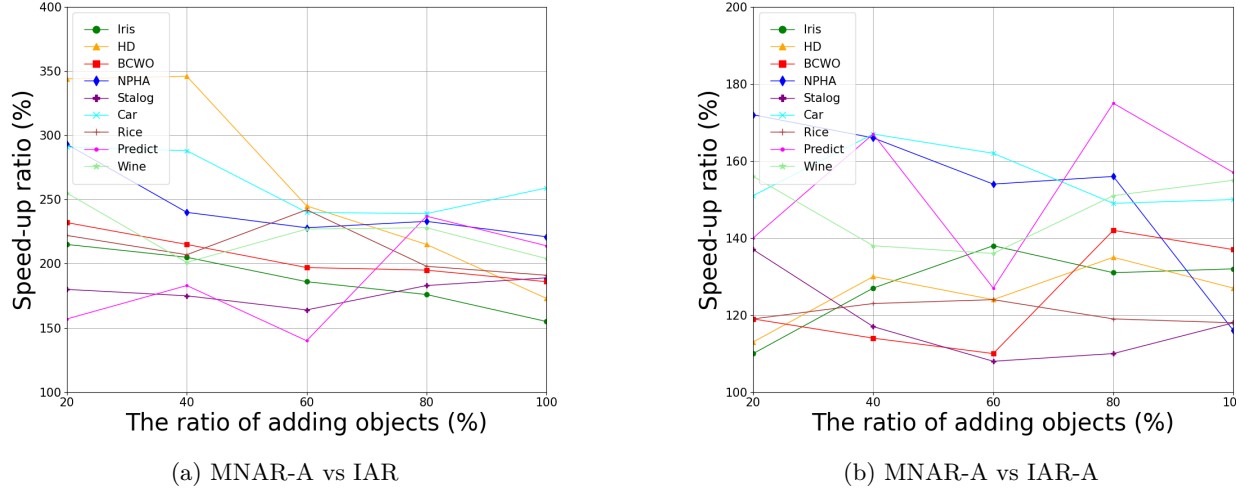

Figure 9: Speedup ratios of MNAR-A over IAR and IAR-A at different object addition ratios across multiple datasets.

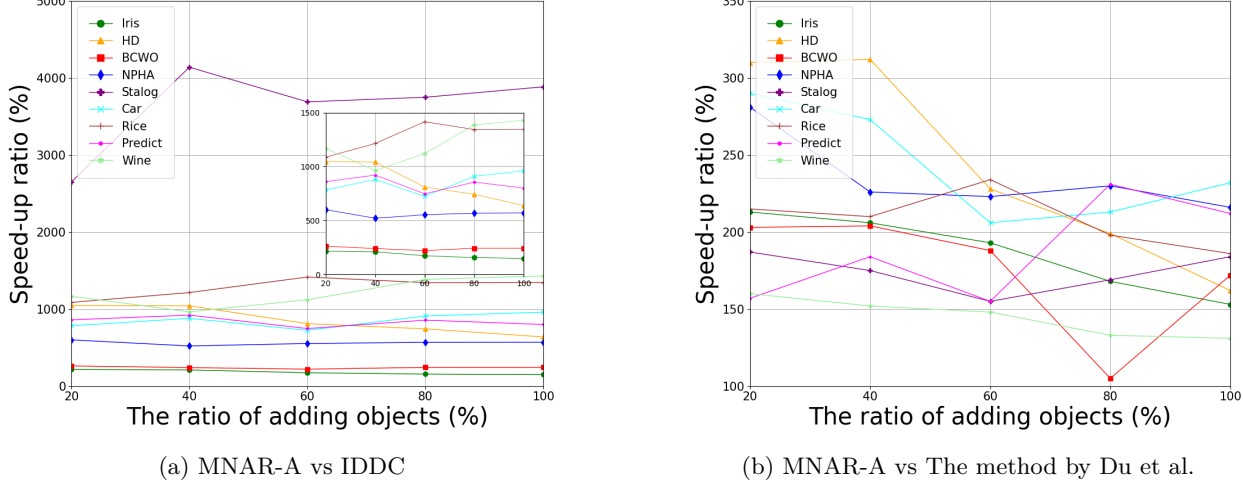

Figure 10: Speedup ratios of MNAR-A over IDDC and The method by Du et al. at different object addition ratios across multiple datasets.

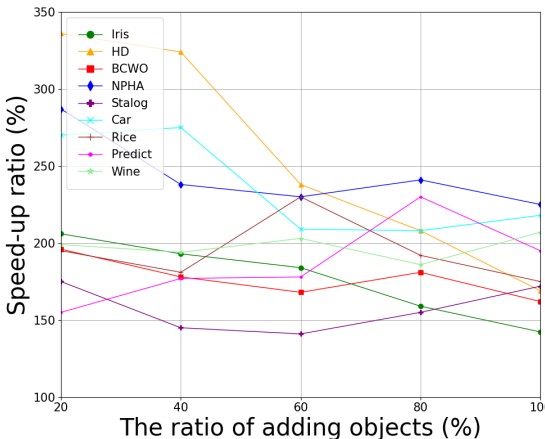

Figure 11: Speedup ratios of MNAR-A over MNAR at different object addition ratios across multiple datasets.

## 6.2 Performance evaluation of MNAR-D

We evaluate the performance of the MNAR-D algorithm in terms of both effectiveness and efficiency. Besides comparing the incremental MNAR-D with its non-incremental counterpart MNAR, we also benchmark it against four existing attribute reduction algorithms: IAR Sang et al. (2021), IAR-D Sang et al. (2021), IDDC Ullah et al. (2024), and the method by Du et al. Du & Hu (2016).

### 6.2.1 Effectiveness

For each dataset, 50% of objects were randomly selected as deleted. Each algorithm computed a new reduction on the remaining data. Classification performance was assessed using SVM, KNN, and RF classifiers with ten-fold cross-validation. Results are presented in Table 18–20 .

MNAR-D compatible or outperformed other methods on most datasets and classifiers, demonstrating its effectiveness in handling object deletions without accuracy loss.

Table 18: Classification accuracy comparison based on KNN (mean ± std).

| Dataset | IAR | IAR-D | IDDC | Du et al. | MNAR | **MNAR-D** |
|---|---|---|---|---|---|---|
| Iris | 1.0000±0.0000 | 1.0000±0.0000 | 1.0000±0.0000 | 0.9589±0.0897 | 1.0000±0.0000 | **1.0000±0.0000** |
| HD | 0.5072±0.0843 | 0.5178±0.0499 | **0.5424±0.0713** | 0.5233±0.0870 | 0.5163±0.0539 | 0.5196±0.0621 |
| BCWO | 0.8824±0.0524 | 0.9443±0.0345 | 0.9542±0.0398 | 0.8303±0.0849 | 0.9541±0.0292 | **0.9571±0.0293** |
| NPHA | 0.5563±0.0600 | 0.5622±0.0350 | 0.5594±0.0410 | 0.5371±0.0742 | 0.5593±0.0642 | **0.5675±0.0576** |
| Statlog | 0.6326±0.0526 | 0.6497±0.0515 | 0.6746±0.0437 | 0.5708±0.1014 | **0.7147±0.0496** | 0.6719±0.0491 |
| Car | 0.8548±0.0452 | 0.8756±0.0524 | 0.8392±0.0846 | 0.7819±0.0902 | 0.8981±0.0711 | **0.9085±0.0590** |
| Rice | 0.9097±0.0219 | 0.9142±0.0226 | 0.9255±0.0210 | 0.9285±0.0368 | 0.9108±0.0214 | **0.9292±0.0218** |
| Predict | 0.5976±0.0136 | 0.6053±0.0202 | 0.5146±0.0749 | 0.5208±0.0739 | 0.6176±0.0404 | **0.6213±0.0297** |
| Wine | 0.6084±0.0278 | **0.6128±0.0368** | 0.4967±0.0111 | 0.5539±0.0970 | 0.5871±0.0275 | 0.5957±0.0207 |
| Average | 0.7277±0.0398 | 0.7424±0.0337 | 0.7327±0.0455 | 0.6895±0.0817 | 0.7509±0.0397 | **0.7576±0.0366** |

Table 19: Classification accuracy comparison based on SVM (mean ± std).

| Dataset | IAR | IAR-D | IDDC | Du et al. | MNAR | **MNAR-D** |
|---|---|---|---|---|---|---|
| Iris | 1.0000±0.0000 | 1.0000±0.0000 | 1.0000±0.0000 | 0.9589±0.0897 | 1.0000±0.0000 | **1.0000±0.0000** |
| HD | 0.5406±0.0204 | 0.5412±0.0108 | 0.5563±0.0299 | 0.5563±0.0299 | 0.5563±0.0299 | **0.5592±0.0289** |
| BCWO | 0.8910±0.0424 | 0.9486±0.0318 | 0.9598±0.0319 | 0.8103±0.1395 | 0.9570±0.0294 | **0.9600±0.0318** |
| NPHA | 0.6069±0.0237 | 0.6211±0.0172 | 0.6126±0.0116 | 0.6126±0.0116 | 0.6126±0.0116 | **0.6274±0.0184** |
| Statlog | 0.5732±0.0254 | 0.5740±0.0266 | 0.5798±0.0633 | 0.5659±0.0874 | 0.5750±0.0669 | **0.5821±0.0612** |
| Car | 0.8848±0.0564 | 0.8895±0.0435 | 0.9225±0.0710 | 0.7917±0.0012 | 0.9225±0.0710 | **0.9225±0.0710** |
| Rice | 0.9192±0.0207 | 0.9206±0.0214 | 0.7145±0.0215 | **0.9217±0.0283** | 0.9166±0.0212 | 0.9166±0.0212 |
| Predict | 0.4993±0.0010 | 0.4993±0.0010 | 0.4762±0.0012 | 0.5335±0.0010 | 0.5335±0.0010 | **0.5335±0.0010** |
| Wine | 0.5139±0.0147 | 0.5420±0.0180 | 0.5198±0.0067 | 0.5120±0.0281 | 0.5198±0.0067 | **0.5476±0.0163** |
| Average | 0.7143±0.0227 | 0.7263±0.0189 | 0.7268±0.0263 | 0.6959±0.0459 | 0.7326±0.0264 | **0.7388±0.0278** |

Table 20: Classification accuracy comparison based on RF (mean ± std).

| Dataset | IAR | IAR-D | IDDC | Du et al. | MNAR | **MNAR-D** |
|---|---|---|---|---|---|---|
| Iris | 1.0000±0.0000 | 1.0000±0.0000 | 1.0000±0.0000 | 1.0000±0.0000 | 1.0000±0.0000 | **1.0000±0.0000** |
| HD | 0.5765±0.0579 | 0.5905±0.0438 | 0.5825±0.0535 | 0.5829±0.0403 | 0.5896±0.0561 | **0.5988±0.0682** |
| BCWO | 0.8853±0.0427 | 0.9400±0.0219 | 0.9570±0.0231 | 0.9398±0.0324 | 0.9426±0.0288 | **0.9586±0.0280** |
| NPHA | 0.5510±0.0682 | **0.6280±0.0143** | 0.5563±0.0833 | 0.5929±0.0320 | 0.6013±0.0171 | 0.5510±0.0837 |
| Statlog | 0.7046±0.0359 | 0.6922±0.0420 | 0.7353±0.0562 | 0.7123±0.0654 | 0.7455±0.0289 | 0.7921±0.0484 |
| Car | 0.7856±0.0739 | 0.8014±0.0069 | 0.9331±0.0724 | 0.7537±0.1446 | 0.8917±0.0012 | **0.9432±0.0695** |
| Rice | 0.9417±0.0121 | 0.9496±0.0213 | 0.9496±0.0213 | 0.9516±0.0271 | 0.9412±0.0196 | **0.9581±0.0108** |
| Predict | 0.7254±0.0164 | 0.7376±0.0131 | 0.7630±0.0186 | 0.7292±0.0175 | 0.7161±0.0186 | **0.7907±0.0249** |
| Wine | 0.6158±0.0326 | 0.6249±0.0494 | 0.6084±0.0426 | 0.5810±0.0273 | 0.6596±0.0228 | **0.6690±0.0377** |
| Average | 0.7540±0.0377 | 0.7738±0.0236 | 0.7872±0.0412 | 0.7604±0.0430 | 0.7875±0.0215 | **0.8068±0.0412** |

### 6.2.2 Sensitivity Experiments on different attribute and object Missing Rates

In order to evaluate how attribute missing rates and sample value missing rates affect the results, we performed experiments using different missing rates on three datasets: BCWO (small attribute count), HD (medium attribute count), and predict (large attribute count).

With the sample missing rate fixed at 10%, we adjusted the attribute missing rate from the original 30% to 40% and 50% to observe the impact. Conversely, while keeping the attribute missing rate constant at 30%, we increased the sample missing rate from the original 10% to 20% and 30%.The results are reported in Table 21–32.

When the sample missing rate and attribute missing rate vary, MNAR-D exhibits only a slight drop in accuracy, and consistently outperforms other methods in terms of classification accuracy under various combinations of sample and attribute missing rates. These results further demonstrate the effectiveness and robustness of our proposed approach.

Table 21: Classification Accuracy Comparison based on KNN under 10% Sample value Missing Rate and 40% Attribute Missing Rate (mean ± std).

| Dataset | IAR | IAR-D | IDDC | DU | MNAR | **MNAR-D** |
|---------|-----|-------|------|----|----|-----------|
| BCWO | 0.9192±0.0328 | 0.9278±0.0341 | 0.9259±0.0227 | 0.7841±0.0298 | 0.9217±0.0297 | **0.9241±0.0309** |
| HD | 0.4738±0.0657 | 0.4765±0.0778 | 0.4879±0.0483 | 0.4867±0.0468 | 0.4852±0.0689 | **0.4971±0.0554** |
| Predict | 0.5449±0.0142 | 0.5561±0.0097 | 0.4615±0.0772 | 0.4685±0.0149 | 0.5733±0.0151 | **0.6258±0.0314** |
| Average | 0.6460±0.0376 | 0.6535±0.0405 | 0.6251±0.0494 | 0.5798±0.0305 | 0.6601±0.0379 | **0.6823±0.0392** |

Table 22: Classification Accuracy Comparison based on KNN under 10% Sample value Missing Rate and 50% Attribute Missing Rate (mean ± std).

| Dataset | IAR | IAR-D | IDDC | DU | MNAR | **MNAR-D** |
|---------|-----|-------|------|----|----|-----------|
| BCWO | 0.8897±0.0339 | 0.8983±0.0353 | 0.8968±0.0239 | 0.7476±0.0319 | 0.8921±0.0318 | **0.9014±0.0331** |
| HD | 0.4472±0.0678 | 0.4500±0.0799 | 0.4628±0.0498 | 0.4619±0.0489 | 0.4595±0.0708 | **0.4734±0.0572** |
| Predict | 0.5104±0.0158 | 0.5200±0.0118 | 0.4364±0.0798 | 0.4330±0.0169 | 0.5397±0.0168 | **0.5991±0.0334** |
| Average | 0.6158±0.0392 | 0.6228±0.0423 | 0.5987±0.0512 | 0.5475±0.0326 | 0.6304±0.0398 | **0.6580±0.0412** |

Table 23: Classification Accuracy Comparison based on KNN under 20% Sample value Missing Rate and 30% Attribute Missing Rate (mean ± std).

| Dataset | IAR | IAR-D | IDDC | DU | MNAR | **MNAR-D** |
|---------|-----|-------|------|----|----|-----------|
| BCWO | 0.9123±0.0329 | 0.9215±0.0344 | 0.9209±0.0229 | 0.7919±0.0299 | 0.9147±0.0298 | **0.9144±0.0301** |
| HD | 0.4584±0.0659 | 0.4610±0.0779 | 0.4758±0.0481 | 0.4749±0.0469 | 0.4703±0.0688 | **0.4893±0.0551** |
| Predict | 0.5493±0.0141 | 0.5599±0.0098 | 0.4699±0.0771 | 0.4766±0.0150 | 0.5793±0.0149 | **0.6258±0.0324** |
| Average | 0.6400±0.0376 | 0.6475±0.0407 | 0.6222±0.0494 | 0.5811±0.0306 | 0.6548±0.0378 | **0.6765±0.0392** |

Table 24: Classification Accuracy Comparison based on SVM under 30% Sample value Missing Rate and 30% Attribute Missing Rate (mean ± std).

| Dataset | IAR | IAR-D | IDDC | DU | MNAR | **MNAR-D** |
|---------|-----|-------|------|----|----|-----------|
| BCWO | 0.9048±0.0338 | 0.9140±0.0354 | 0.9135±0.0238 | 0.7847±0.0318 | 0.9071±0.0319 | **0.9091±0.0312** |
| HD | 0.4509±0.0679 | 0.4535±0.0798 | 0.4691±0.0499 | 0.4683±0.0488 | 0.4629±0.0709 | **0.4840±0.0571** |
| Predict | 0.5426±0.0159 | 0.5523±0.0117 | 0.4632±0.0799 | 0.4691±0.0168 | 0.5709±0.0169 | **0.6201±0.0344** |
| Average | 0.6328±0.0392 | 0.6399±0.0423 | 0.6153±0.0512 | 0.5740±0.0325 | 0.6470±0.0399 | **0.6711±0.0409** |

Table 25: Classification Accuracy Comparison based on SVM under 10% Sample value Missing Rate and 40% Attribute Missing Rate (mean ± std).

| Dataset | IAR | IAR-D | IDDC | DU | MNAR | **MNAR-D** |
|---------|-----|-------|------|----|----|-----------|
| BCWO | 0.8938±0.0289 | 0.8915±0.0311 | 0.9007±0.0321 | 0.8212±0.0151 | 0.8964±0.0231 | **0.9150±0.0241** |
| HD | 0.4591±0.0414 | 0.4910±0.0407 | 0.4925±0.0150 | 0.4915±0.0112 | 0.4925±0.0130 | **0.5116±0.0114** |
| Predict | 0.4416±0.0262 | 0.4416±0.0220 | 0.4060±0.0433 | 0.4366±0.0322 | 0.4416±0.0224 | **0.4858±0.0211** |
| Average | 0.5982±0.0322 | 0.6080±0.0313 | 0.5997±0.0301 | 0.5831±0.0195 | 0.6102±0.0195 | **0.6375±0.0189** |

Table 26: Classification Accuracy Comparison based on SVM under 10% Sample value Missing Rate and 50% Attribute Missing Rate (mean ± std).

| Dataset | IAR | IAR-D | IDDC | DU | MNAR | **MNAR-D** |
|---------|-----|-------|------|-----|------|------------|
| BCWO | 0.8658±0.0301 | 0.8635±0.0321 | 0.8727±0.0331 | 0.7862±0.0171 | 0.8684±0.0251 | **0.8900±0.0261** |
| HD | 0.4291±0.0115 | 0.4610±0.0468 | 0.4625±0.0313 | 0.4615±0.0013 | 0.4625±0.0326 | **0.4866±0.0315** |
| Predict | 0.4116±0.0353 | 0.4116±0.0303 | 0.3760±0.0204 | 0.4066±0.0400 | 0.4116±0.0223 | **0.4608±0.0231** |
| Average | 0.5688±0.0256 | 0.5787±0.0364 | 0.5704±0.0283 | 0.5514±0.0195 | 0.5808±0.0267 | **0.6125±0.0269** |

Table 27: Classification Accuracy Comparison based on SVM under 20% Sample value Missing Rate and 30% Attribute Missing Rate (mean ± std).

| Dataset | IAR | IAR-D | IDDC | DU | MNAR | **MNAR-D** |
|---------|-----|-------|------|-----|------|------------|
| BCWO | 0.9218±0.0281 | 0.9195±0.0301 | 0.9287±0.0311 | 0.8492±0.0141 | 0.9244±0.0221 | **0.9400±0.0231** |
| HD | 0.4791±0.0323 | 0.5210±0.0236 | 0.5225±0.0147 | 0.5215±0.0026 | 0.5225±0.0255 | **0.5356±0.0230** |
| Predict | 0.4666±0.0423 | 0.4666±0.0266 | 0.4360±0.0225 | 0.4666±0.0320 | 0.4666±0.0200 | **0.5108±0.0126** |
| Average | 0.6225±0.0342 | 0.6357±0.0268 | 0.6291±0.0228 | 0.6124±0.0162 | 0.6378±0.0225 | **0.6621±0.0196** |

Table 28: Classification Accuracy Comparison based on SVM under 30% Sample value Missing Rate and 30% Attribute Missing Rate (mean ± std).

| Dataset | IAR | IAR-D | IDDC | DU | MNAR | **MNAR-D** |
|---------|-----|-------|------|-----|------|------------|
| BCWO | 0.9158±0.0291 | 0.9135±0.0311 | 0.9227±0.0321 | 0.8432±0.0151 | 0.9184±0.0231 | **0.9370±0.0241** |
| HD | 0.4731±0.0114 | 0.5150±0.0217 | 0.5165±0.0112 | 0.5155±0.0312 | 0.5165±0.0256 | **0.5326±0.0326** |
| Predict | 0.4606±0.0288 | 0.4606±0.0322 | 0.4300±0.0326 | 0.4606±0.0222 | 0.4606±0.0811 | **0.5078±0.0321** |
| Average | 0.6165±0.0231 | 0.6297±0.0283 | 0.6231±0.0253 | 0.6064±0.0228 | 0.6318±0.0433 | **0.6591±0.0296** |

Table 29: Classification Accuracy Comparison based on RF under 10% Sample value Missing Rate and 40% Attribute Missing Rate (mean ± std).

| Dataset | IAR | IAR-D | IDDC | DU | MNAR | **MNAR-D** |
|---------|-----|-------|------|-----|------|------------|
| BCWO | 0.8998±0.0358 | 0.8984±0.0328 | 0.9126±0.0274 | 0.9112±0.0274 | 0.9055±0.0214 | **0.9359±0.0174** |
| HD | 0.5471±0.0298 | 0.5474±0.0368 | 0.5529±0.0608 | 0.5798±0.0498 | 0.5167±0.0468 | **0.5820±0.0398** |
| Predict | 0.6332±0.0098 | 0.6956±0.0088 | 0.6974±0.0108 | 0.6533±0.0118 | 0.6605±0.0098 | **0.7307±0.0078** |
| Average | 0.6934±0.0251 | 0.7138±0.0261 | 0.7210±0.0330 | 0.7148±0.0297 | 0.6942±0.0260 | **0.7495±0.0217** |

Table 30: Classification Accuracy Comparison based on RF under 10% Sample value Missing Rate and 50% Attribute Missing Rate (mean ± std).

| Dataset | IAR | IAR-D | IDDC | DU | MNAR | **MNAR-D** |
|---------|-----|-------|------|-----|------|------------|
| BCWO | 0.8746±0.0374 | 0.8732±0.0344 | 0.8875±0.0284 | 0.8861±0.0284 | 0.8803±0.0224 | **0.9159±0.0184** |
| HD | 0.5271±0.0308 | 0.5274±0.0378 | 0.5329±0.0628 | 0.5598±0.0508 | 0.4967±0.0488 | **0.5620±0.0418** |
| Predict | 0.6032±0.0108 | 0.6656±0.0098 | 0.6674±0.0118 | 0.6233±0.0128 | 0.6305±0.0108 | **0.7057±0.0088** |
| Average | 0.6683±0.0263 | 0.6887±0.0273 | 0.6959±0.0343 | 0.6897±0.0307 | 0.6692±0.0273 | **0.7279±0.0230** |

Table 31: Classification Accuracy Comparison based on RF under 20% Sample value Missing Rate and 30% Attribute Missing Rate (mean ± std).

| Dataset | IAR | IAR-D | IDDC | DU | MNAR | **MNAR-D** |
|---|---|---|---|---|---|---|
| BCWO | 0.9196±0.0354 | 0.9182±0.0324 | 0.9325±0.0264 | 0.9311±0.0264 | 0.9253±0.0204 | **0.9529±0.0164** |
| HD | 0.5671±0.0288 | 0.5674±0.0358 | 0.5729±0.0598 | 0.5948±0.0488 | 0.5367±0.0458 | **0.6099±0.0388** |
| Predict | 0.6582±0.0088 | 0.7206±0.0078 | 0.7224±0.0098 | 0.6783±0.0108 | 0.6755±0.0088 | **0.7537±0.0068** |
| Average | 0.7150±0.0243 | 0.7354±0.0253 | 0.7426±0.0320 | 0.7347±0.0287 | 0.7125±0.0250 | **0.7722±0.0207** |

Table 32: Classification Accuracy Comparison based on RF under 30% Sample value Missing Rate and 30% Attribute Missing Rate (mean ± std).

| Dataset | IAR | IAR-D | IDDC | DU | MNAR | **MNAR-D** |
|---|---|---|---|---|---|---|
| BCWO | 0.9126±0.0364 | 0.9112±0.0334 | 0.9255±0.0274 | 0.9241±0.0274 | 0.9183±0.0214 | **0.9489±0.0174** |
| HD | 0.5601±0.0298 | 0.5604±0.0368 | 0.5659±0.0608 | 0.5878±0.0498 | 0.5297±0.0468 | **0.6039±0.0398** |
| Predict | 0.6512±0.0098 | 0.7136±0.0088 | 0.7154±0.0108 | 0.6713±0.0118 | 0.6685±0.0098 | **0.7477±0.0078** |
| Average | 0.7080±0.0253 | 0.7284±0.0263 | 0.7356±0.0330 | 0.7277±0.0297 | 0.7055±0.0260 | **0.7668±0.0217** |

### 6.2.3 Efficiency

For each dataset, we first randomly marked 50% of the objects for deletion. We then progressively removed 20%, 40%, 60%, 80%, and finally 100% of these marked objects, measuring the runtime of MNAR-D and the baseline algorithms after each step. Figure 12- 19plots the resulting runtime curves.

As object deletion proportion increased, running times of all algorithms decreased. MNAR-D consistently required less time due to more efficient dominance matrix computations. IDDC remained the slowest due to complex dependency calculations.

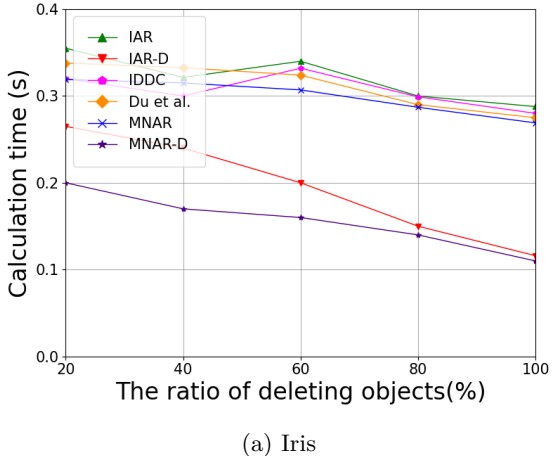
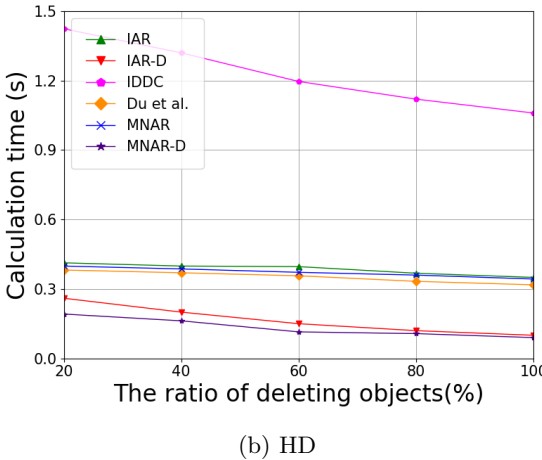

(a) Iris

(b) HD

Figure 12: Computation time on Iris and HD datasets at varying deleted object ratios.

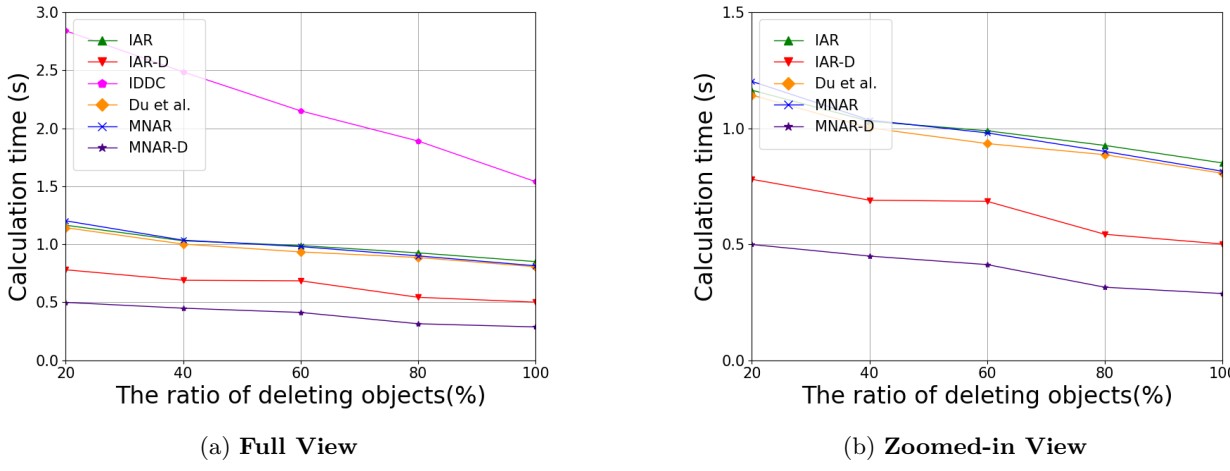

(a) **Full View**          (b) **Zoomed-in View**

Figure 13: Computation time on the **NPHA dataset** at varying added object ratios. (**Right**: Zoomed-in view of the $0 - 1.5$ s range to distinguish lower-running-time methods.)

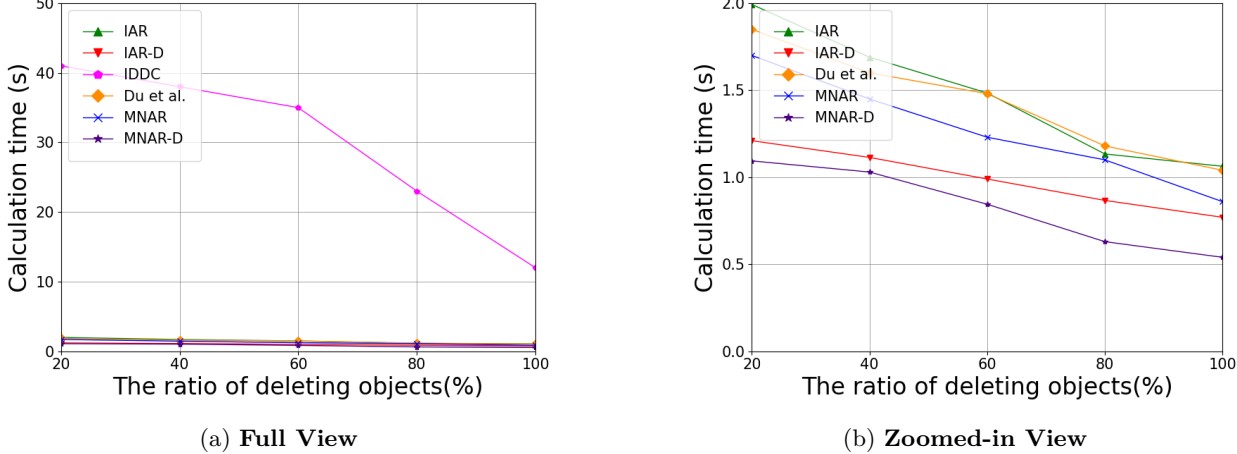

(a) **Full View**          (b) **Zoomed-in View**

Figure 14: Computation time on the **Stalog dataset** at varying added object ratios. (**Right**: Zoomed-in view of the $0 - 2.0$ s range to distinguish lower-running-time methods.)

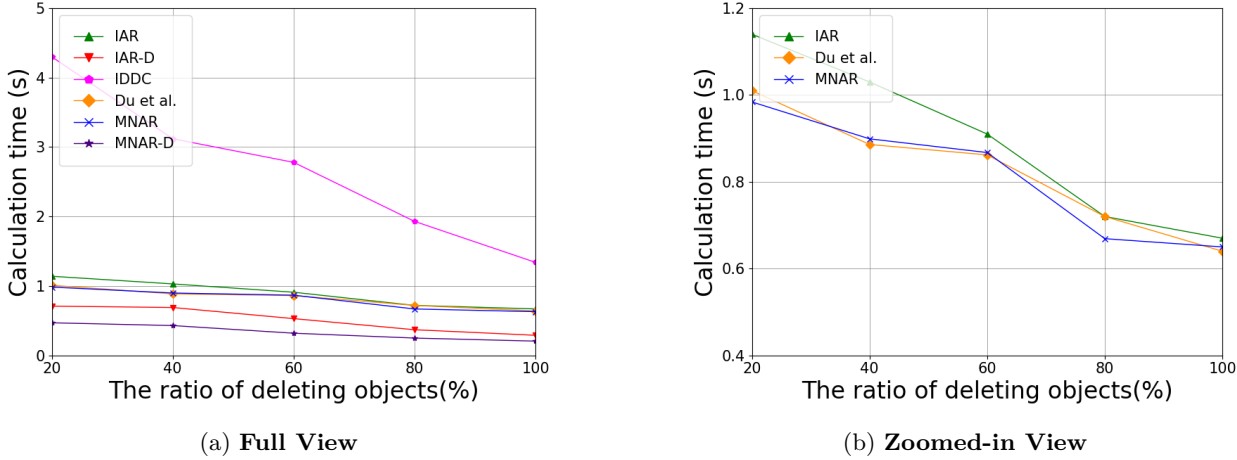

(a) **Full View**          (b) **Zoomed-in View**

Figure 15: Computation time on the **Car dataset** at varying added object ratios. (**Right**: Zoomed-in view of the $0 - 1.2$ s range to distinguish lower-running-time methods.)

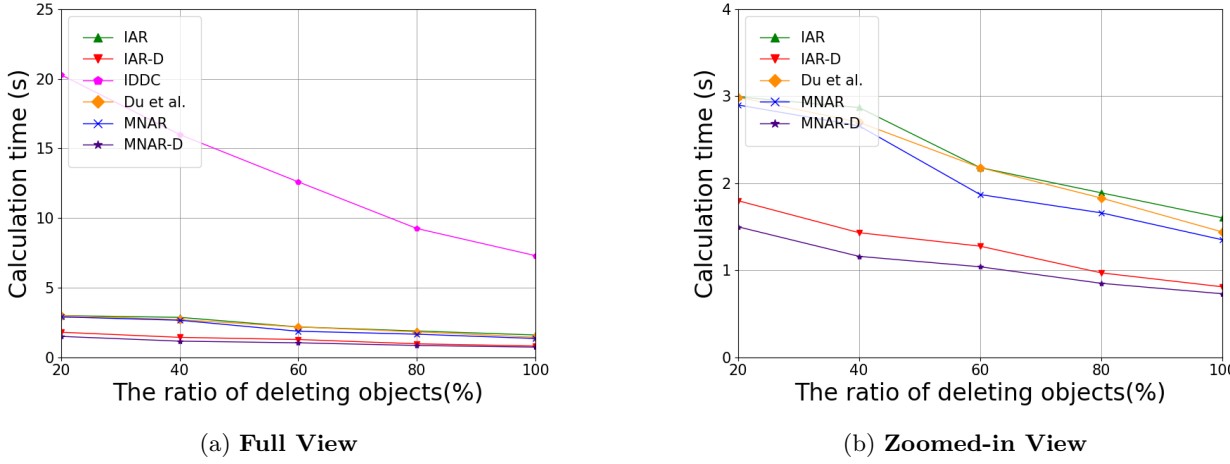

(a) **Full View**

(b) **Zoomed-in View**

Figure 16: Computation time on the **Rice dataset** at varying added object ratios. (**Right**: Zoomed-in view of the $0 - 4.0$ s range to distinguish lower-running-time methods.)

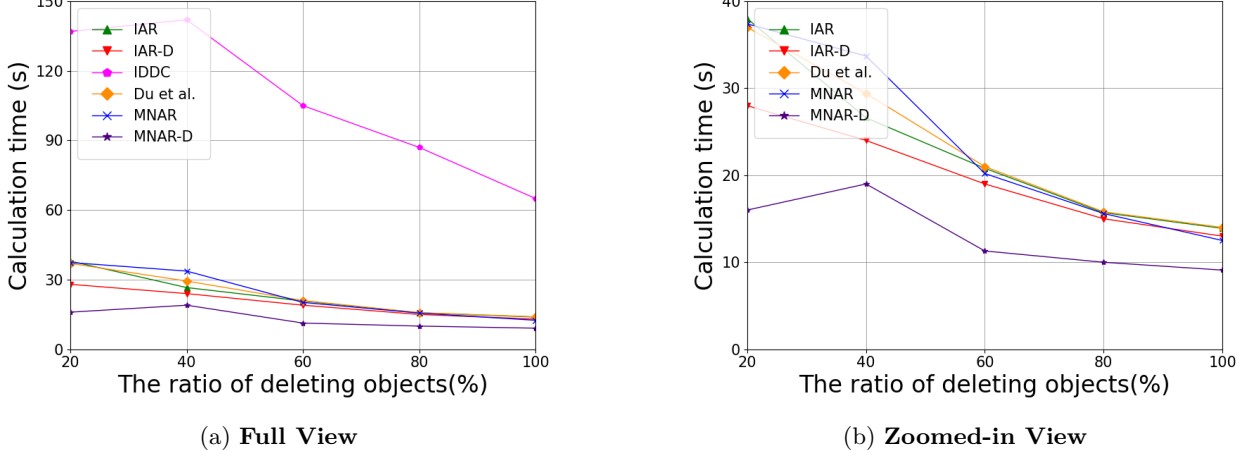

(a) **Full View**

(b) **Zoomed-in View**

Figure 17: Computation time on the **Predict dataset** at varying added object ratios. (**Right**: Zoomed-in view of the $0 - 40$ s range to distinguish lower-running-time methods.)

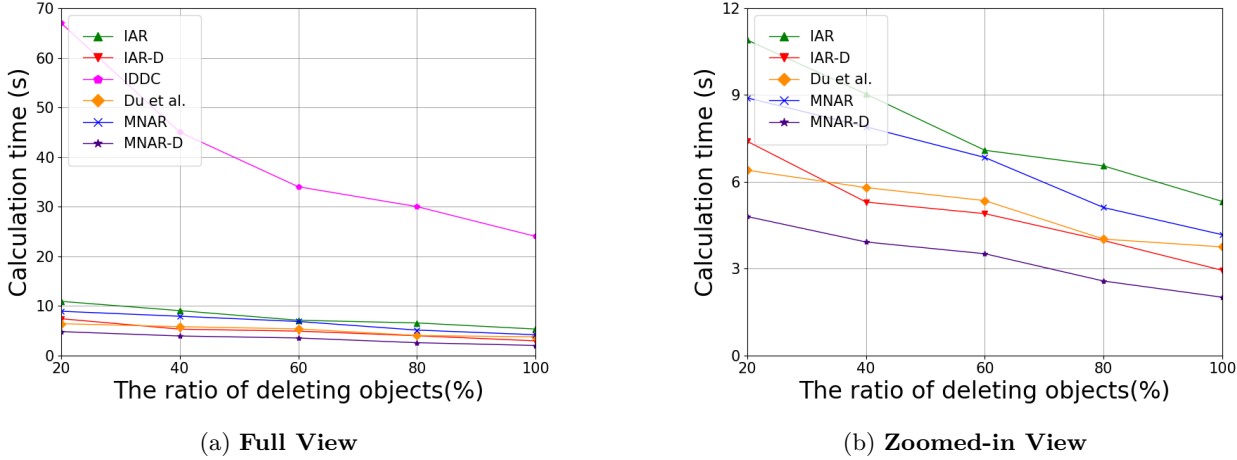

(a) **Full View**

(b) **Zoomed-in View**

Figure 18: Computation time on the **Wine dataset** at varying added object ratios. (**Right**: Zoomed-in view of the $0 - 12$ s range to distinguish lower-running-time methods.)

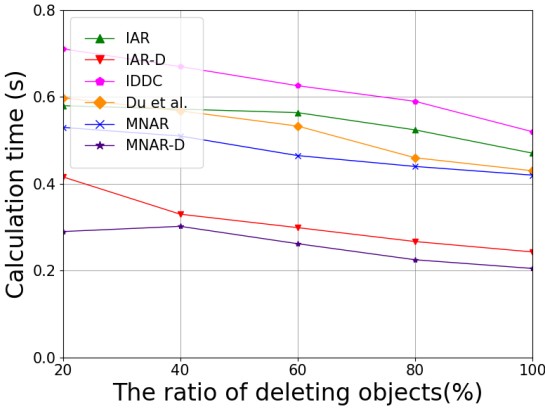

Figure 19: Computation time on BCWO dataset at varying deleted object ratios.

Figure 20- 22 further illustrates the speedup ratios of MNAR-D over the competing methods across different object deletion stages on various dataset. The speedup is defined as the ratio between the runtime of each baseline method and that of MNAR-D under the same deleted-object ratio.

As shown in the figure, MNAR-D consistently achieves speedup values greater than 100% on all datasets and at all insertion stages, indicating a stable and persistent efficiency advantage throughout the incremental process. Although the absolute magnitude of the speedup varies across datasets, the overall trends remain consistent, without noticeable degradation as more objects are deleted.

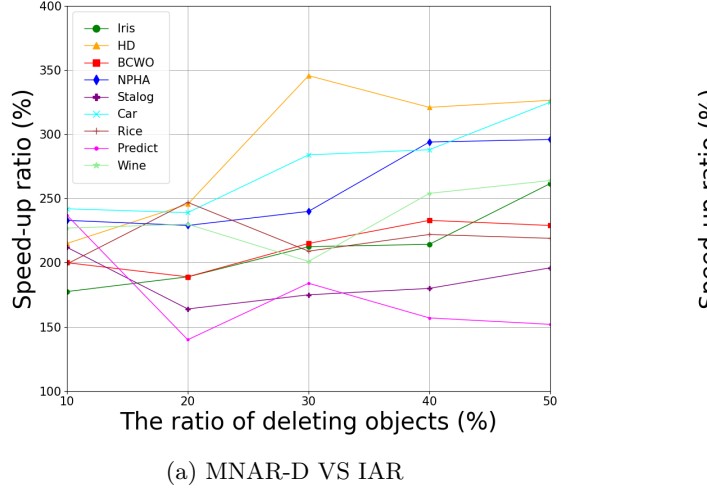

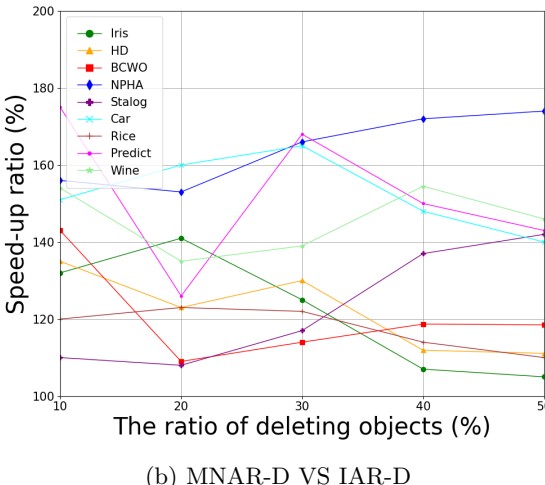

(a) MNAR-D VS IAR

(b) MNAR-D VS IAR-D

Figure 20: Speedup ratios of MNAR-D over IAR and IAR-D at different object deletion ratios across multiple datasets.

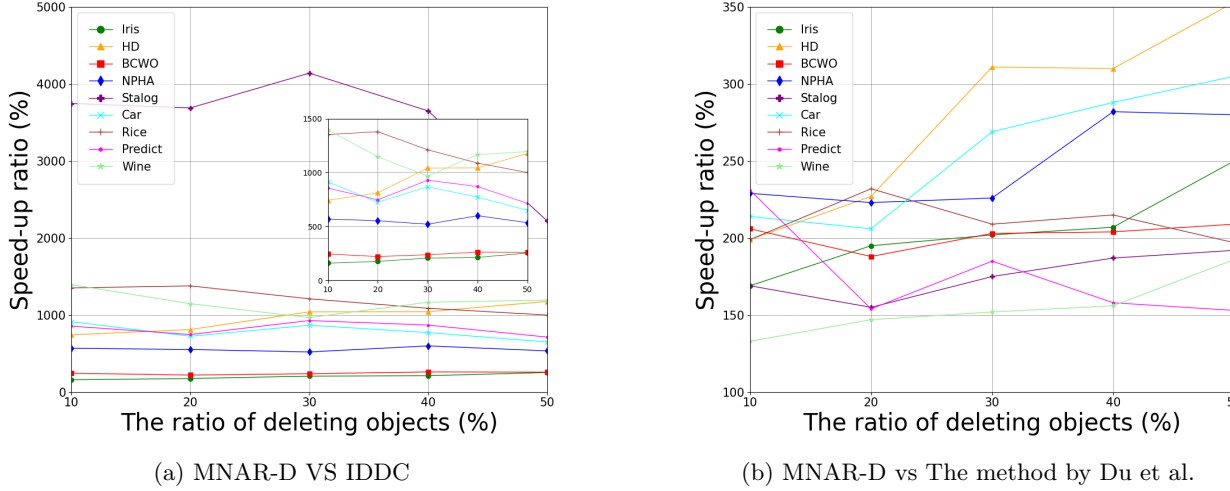

(a) MNAR-D VS IDDC            (b) MNAR-D vs The method by Du et al.

Figure 21: Speedup ratios of MNAR-D over IDDC and The method by Du et al. at different object deletion ratios across multiple datasets.

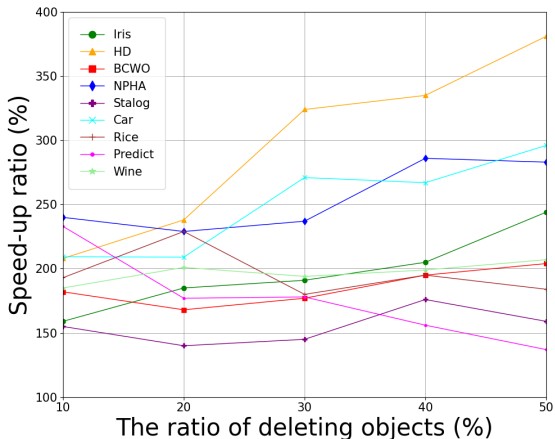

Figure 22: Speedup ratios of MNAR-D over MNAR at different object deletion ratios across multiple datasets.

## 7 Conclusion

h In this study, we present novel incremental attribute reduction algorithms to handle dynamically changing data in IODS. First, we develop two formulations of the CDS for IODS: a set-based definition and a matrix-based counterpart, and adopt the resulting GDS as our attribute-importance metric. Second, we derive incremental update rules for matrix-based GDS and, on this foundation, design wo incremental attribute reduction algorithms: MNAR-A and MNAR-D. Experimental results on UCI datasets showed that the proposed approaches achieved a $1.3\times$ speedup and delivered a 7% relative accuracy gain compared to the SOTA method on average, which demonstrates the effectiveness of our approaches.

In the era of big data, the changes in IODS are diverse. In addition to changes in data sets, modifications to attribute sets and attribute values also occur. The algorithms proposed in this paper are not yet applicable to these attribute changes. Therefore, in future research, we will study the dynamic changes of attribute sets and attribute values in IODS separately and develop incremental methods for attribute reduction.

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

# A Appendix

You may include other additional sections here.

