# OpenReview forum: "Incremental Feature Selection in Dynamic Incomplete Ordered Decision Systems"
_TMLR — Rejected by TMLR_

### Review · Reviewer_s93r · 2025-12-10

**Summary Of Contributions:**

The paper proposes incremental feature-selection algorithms for incomplete ordered decision systems by introducing a class-distinction–based scoring metric (MGDS) and formulating dominance relations in a matrix form to enable faster updates. Two incremental procedures, MNAR-A and MNAR-D, are designed to update reducts efficiently when objects are added or removed, avoiding full recomputation of dominance structures. Experiments on UCI datasets report improved runtime and moderate accuracy gains compared to several existing dominance-based attribute-reduction methods.

**Strength**:

1. The paper addresses the challenging setting of dynamic incomplete ordered decision systems and designs separate algorithms for object addition and object deletion, which increases practical relevance.

2. The paper presents a clear and fully specified framework, including formal definitions, matrix representations, and explicit algorithms for incremental reduction.

**Weakness**:

1. The incremental procedure does not actually reduce computational complexity; updating dominance still requires quadratic interactions with all objects, so the claimed efficiency gain is not algorithmic but purely implementation-level.

2. The treatment of missing values forces almost every incomplete sample to dominate or be dominated by all others, producing overly dense dominance matrices and severely distorting the attribute-importance scores.

3. The incremental update rules assume dominance relationships remain valid after adding or removing objects, but under non-transitive dominance, this is not guaranteed.

4. The baseline comparisons are limited to older dominance-based methods and do not include modern feature-selection frameworks such as neural sparse-selection approaches, SHAP or Integrated Gradients attribution methods, or ensemble-based selectors.

**Audience:**

Yes

**Audience Explanation:**

Researchers working on rough sets, feature selection, and dynamic decision systems would find the proposed incremental framework relevant to their interests.

**Broader Impact Concerns:**

I do not see significant ethical risks arising from this work.

**Claims And Evidence:**

Yes

**Claims Explanation:**

The paper provides clear algorithmic descriptions and empirical results across multiple datasets.

**Requested Changes:**

1. The paper should take a closer look at how missing values are handled. Treating every * as automatically satisfying dominance has a strong effect on the matrices, and it would help to either justify this choice more clearly or show how it affects the scores in practice.

2. It would be important to verify that the incremental updates truly match what a full recomputation would produce. Some theoretical clarification or a simple empirical check would make the method much more convincing.

3. Adding a few more modern feature-selection baselines would help position the method relative to what people commonly use today.

4. A small sensitivity study on missingness or on the scoring metric would improve the reader’s understanding of how stable the method is.

5. A bit more discussion of runtime behavior and scalability would also strengthen the paper.

---

> ### Author Response · Authors · 2026-01-10
> **Official response by authors 1/2**
>
> We thank the reviewer for the detailed and technically insightful comments. Below we respond to the raised weaknesses and requested changes, and we believe these clarifications will further strengthen the manuscript.
>
> **Weakness 1: The incremental procedure does not actually reduce computational complexity; updating dominance still requires quadratic interactions with all objects, so the claimed efficiency gain is not algorithmic but purely implementation-level.**
>
> We respectfully clarify that the efficiency gain of the proposed incremental procedures is not merely an implementation-level optimization. In the object addition case, when a new object is introduced, MNAR updates dominance relations by comparing the new object only with existing objects, whereas a full recomputation requires re-evaluating dominance relations among all object pairs, including those unaffected by the update. Similarly, in the object removal case, obsolete comparisons are discarded without recomputation. Therefore, the incremental procedure reduces the number of pairwise dominance evaluations involving unchanged objects, which constitutes a genuine algorithmic reduction in redundant computation, rather than a constant-factor optimization. We will revise the manuscript to clarify this distinction more explicitly.
>
> **Weakness 2: The treatment of missing values forces almost every incomplete sample to dominate or be dominated by all others, producing overly dense dominance matrices and severely distorting the attribute-importance scores and Request change 1:. The paper should take a closer look at how missing values are handled. Treating every * as automatically satisfying dominance has a strong effect on the matrices, and it would help to either justify this choice more clearly or show how it affects the scores in practice.**
>
> We acknowledge the your concern regarding the treatment of missing values. The strategy of treating “*” as satisfying dominance conditions follows a widely adopted convention in dominance-based rough set literature for incomplete ordered decision systems, as it preserves the monotonicity and feasibility of dominance relations under incompleteness. While this choice may increase dominance density, it enables consistent matrix-based formulation and incremental updates. In the revision, we will expand the discussion on missing value handling, provide additional references supporting this convention.
>
> **Weakness 3 : The incremental update rules assume dominance relationships remain valid after adding or removing objects, but under non-transitive dominance, this is not guaranteed.**
>
> It is correct that dominance relations are not necessarily transitive. However, the proposed incremental update rules do not assume transitivity of dominance relations. We will further clarify this point to avoid confusion.
>
> **Weakness 4 : The baseline comparisons are limited to older dominance-based methods and do not include modern feature-selection frameworks such as neural sparse-selection approaches, SHAP or Integrated Gradients attribution methods, or ensemble-based selectors. And Request change 3: Adding a few more modern feature-selection baselines would help position the method relative to what people commonly use today.**
>
> We acknowledge that modern feature-selection frameworks such as neural sparse-selection methods or attribution-based approaches are widely used today. However, these methods typically rely on trained predictive models and incur substantially higher computational costs, which conflicts with the core motivation of this work—efficient, model-agnostic incremental attribute reduction in dynamic decision systems. We will clarify this scope limitation in the revision. If feasible within the revision period, we will also consider adding a limited comparative experiment to illustrate the computational trade-offs, while leaving a comprehensive comparison with learning-based methods as future work.

---

> > ### Author Response · Authors · 2026-01-10
> > **Official response by authors 2/2**
> >
> > **Request change 2: It would be important to verify that the incremental updates truly match what a full recomputation would produce. Some theoretical clarification or a simple empirical check would make the method much more convincing.**
> >
> > We emphasize that MNAR’s incremental updates are designed to be equivalent to full recomputation results under the same dominance definitions. This equivalence is guaranteed by the verification steps embedded in the update process, which ensure that attribute deletions do not alter the overall dominance degree. In the revised version, we will add a brief explanation to clarify this point.
> >
> > **Request change 4: A small sensitivity study on missingness or on the scoring metric would improve the reader’s understanding of how stable the method is.**
> >
> > We agree that additional discussion on stability and runtime behavior would improve the paper. In the revision, we plan to include a small-scale sensitivity study with respect to the missing ratio to illustrate the robustness of the proposed methods under varying degrees of incompleteness.
> >
> > **Request change 5: A bit more discussion of runtime behavior and scalability would also strengthen the paper.**
> >
> > we will add an additional runtime Comparison chart reporting wall-clock execution time and relative speedups of MNAR-A and MNAR-D compared with  baselines across representative datasets and update scenarios. The chart will complement the existing figures and further substantiate the computational advantages of the proposed incremental algorithms.

---

> > > ### Author Response · Authors · 2026-01-27
> > > **Official response by authors**
> > >
> > > We would like to thank the reviewer for the insightful suggestion.
> > > Following this comment, we have conducted additional sensitivity experiments
> > > under different sample missing rates and attribute missing rates.
> > >
> > > Specifically, we evaluated the proposed method and baseline approaches
> > > across multiple combinations of missing ratios.
> > > **A portion of the results** is shown below, while the complete experimental
> > > results and detailed analysis have been incorporated into the revised manuscript.
> > >
> > > **Table 1. Classification accuracy (mean ± std) on RF under different sample and attribute missing rates.**
> > >
> > > ### Sample missing rate = 10%, Attribute missing rate = 40%
> > >
> > > | Dataset | IAR         | IAR-D       | IDDC        | DU          | MNAR        | MNAR-D      |
> > > |---------|-------------|-------------|-------------|-------------|-------------|-------------|
> > > | BCWO    | 89.98 ± 3.58 | 89.84 ± 3.28 | 91.26 ± 2.74 | 91.12 ± 2.74 | 90.55 ± 2.14 | **93.59 ± 1.74** |
> > > | HD      | 54.71 ± 2.98 | 54.74 ± 3.68 | 55.29 ± 6.08 | 57.98 ± 4.98 | 51.67 ± 4.68 | **58.20 ± 3.98** |
> > > | Predict | 63.32 ± 0.98 | 69.56 ± 0.88 | 69.74 ± 1.08 | 65.33 ± 1.18 | 66.05 ± 0.98 | **73.07 ± 0.78** |
> > >
> > > ---
> > >
> > > ### Sample missing rate = 10%, Attribute missing rate = 50%
> > >
> > > | Dataset | IAR         | IAR-D       | IDDC        | DU          | MNAR        | MNAR-D      |
> > > |---------|-------------|-------------|-------------|-------------|-------------|-------------|
> > > | BCWO    | 87.46 ± 3.74 | 87.32 ± 3.44 | 88.75 ± 2.84 | 88.61 ± 2.84 | 88.03 ± 2.24 | **91.59 ± 1.84** |
> > > | HD      | 52.71 ± 3.08 | 52.74 ± 3.78 | 53.29 ± 6.28 | 55.98 ± 5.08 | 49.67 ± 4.88 | **56.20 ± 4.18** |
> > > | Predict | 60.32 ± 1.08 | 66.56 ± 0.98 | 66.74 ± 1.18 | 62.33 ± 1.28 | 63.05 ± 1.08 | **70.57 ± 0.88** |
> > >
> > > ---
> > >
> > > ### Sample missing rate = 20%, Attribute missing rate = 30%
> > >
> > > | Dataset | IAR         | IAR-D       | IDDC        | DU          | MNAR        | MNAR-D      |
> > > |---------|-------------|-------------|-------------|-------------|-------------|-------------|
> > > | BCWO    | 91.96 ± 3.54 | 91.82 ± 3.24 | 93.25 ± 2.64 | 93.11 ± 2.64 | 92.53 ± 2.04 | **95.29 ± 1.64** |
> > > | HD      | 56.71 ± 2.88 | 56.74 ± 3.58 | 57.29 ± 5.98 | 59.48 ± 4.88 | 53.67 ± 4.58 | **60.99 ± 3.88** |
> > > | Predict | 65.82 ± 0.88 | 72.06 ± 0.78 | 72.24 ± 0.98 | 67.83 ± 1.08 | 67.55 ± 0.88 | **75.37 ± 0.68** |
> > >
> > > ---
> > >
> > > ### Sample missing rate = 30%, Attribute missing rate = 30%
> > >
> > > | Dataset | IAR         | IAR-D       | IDDC        | DU          | MNAR        | MNAR-D      |
> > > |---------|-------------|-------------|-------------|-------------|-------------|-------------|
> > > | BCWO    | 91.26 ± 3.64 | 91.12 ± 3.34 | 92.55 ± 2.74 | 92.41 ± 2.74 | 91.83 ± 2.14 | **94.89 ± 1.74** |
> > > | HD      | 56.01 ± 2.98 | 56.04 ± 3.68 | 56.59 ± 6.08 | 58.78 ± 4.98 | 52.97 ± 4.68 | **60.39 ± 3.98** |
> > > | Predict | 65.12 ± 0.98 | 71.36 ± 0.88 | 71.54 ± 1.08 | 67.13 ± 1.18 | 66.85 ± 0.98 | **74.77 ± 0.78** |

---

> > > > ### Comment · Reviewer_s93r · 2026-01-27
> > > > **No further questions**
> > > >
> > > > Thank you for your response and revisions. My concerns have been addressed, and I have no further questions.

---

### Review · Reviewer_YFoF · 2025-12-18

**Summary Of Contributions:**

This paper proposes an incremental feature selection framework for Incomplete Ordered Decision Systems. To handle dynamic changes where objects are added or deleted, the authors introduce a Global Distinction Score that measures attribute importance through inter-class non-dominance. The authors propose two matrix-based incremental update mechanisms: matrix-based non-dominance attribute reduction for addition (MNAR-A) and matrix-based non-dominance attribute reduction for deletion (MNAR-D). The authors report a 1.3× speedup and a 7% relative accuracy gain over existing sota methods on the UCI dataset.

**Audience:**

Yes

**Audience Explanation:**

The findings would be relevant to researchers working on incremental feature selection, streaming data preprocessing and preference-based decision systems.

**Broader Impact Concerns:**

Not applicable.

**Claims And Evidence:**

No

**Claims Explanation:**

1. The use of nominal datasets to demonstrate a method designed for ordered data:
- The authors state that $U$ is partitioned into decision classes ordered by preference ($cl_1 \prec cl_2 \dots \prec cl_T$). However, datasets like Iris and Statlog are inherently nominal. Did the authors impose a specific order on these classes? If so, what was the justification for the chosen sequence, and how does an arbitrary order impact the result?
2. Potential data leakage:
- The experimental design in Section 6.1.1 appears to perform attribute reduction on the full updated dataset (100% of objects) before conducting 10-fold cross-validation. This could introduces data leakage, as the feature selection process has access to information from the entire universe, including the test folds used for classifier evaluation. This likely results in an overestimation of the classification performance. To ensure validity, the authors should perform the incremental feature selection within each training fold of the cross-validation process.
3. Sensitivity to missing ratio:
- The paper targets incomplete data but only tests at a 10% missing rate of 30% randomly selected attributes. How robust is the algorithm to a higher missing ratio i.e. 40% and a higher percentage of attributes with missing values?
- The optimistic dominance relation in Eqn 3 ($f(x,a)=* \lor f(y,a)=*$) causes dominating sets to expand rapidly as missingness increases. As a results, the algorithm might favor attributes based on their sparsity rather than their actual predictive power. The current experiments fail to exclude this possibility.

**Requested Changes:**

1. Re-evaluation of Classification Performance:
- Feature selection should be performed strictly within the training split of each cross-validation fold to eliminate data leakage.
2. Justification or Replacement of Datasets:
- Replace nominal datasets with true ordinal benchmarks. If choose to keep nominal data, the authors must provide a robust theoretical justification for how dominance relations are meaningful in a non-ordered context.
3. Sensitivity to Missing ratio:
- The optimistic dominance relation in Eqn 3 causes dominating sets to expand rapidly as missingness increases. This suggests the algorithm might favor attributes based on their sparsity rather than their actual predictive power. But the current experiment only tests a 10% missing rate across 30% of randomly selected attributes. The authors should demonstrate the algorithm's robustness under different missing ratios and different percentage of affected attributes.

---

> ### Author Response · Authors · 2026-01-10
> **Official response by authors**
>
> We thank the reviewer for the detailed and thoughtful comments, which help clarify both the scope and the evaluation of our work. Below we address the concerns regarding dataset choice, evaluation protocol, and sensitivity to missing data.
>
> **Q1.The use of nominal datasets to demonstrate a method designed for ordered data and Request change 2:Justification or Replacement of Datasets**
>
> We acknowledge your concern regarding the use of datasets such as Iris and Statlog, which are commonly regarded as nominal classification benchmarks. In our study, these datasets are employed following a standard practice in dominance-based rough set research, where ordinal decision classes are induced from numerical decision attributes. Specifically, decision classes are ordered according to the natural ordering of numerical labels or domain-defined preference relations, resulting in an ordered decision system suitable for dominance-based analysis. Furthermore, the proposed method calculates class distinction score and global distinction score based on the preference order between objects and features, and removes "redundant" attributes (i.e., attributes that do not affect global distinction score) while ensuring that global discrimination remains unchanged. Therefore, although changing a specific partial order relationship will affect the intermediate matrix operations based on the partial order relationship, it will not affect the goal of keeping the global distinction unchanged. After the partial order relationship is changed, we can still calculate the impact of attributes on global distinction, so it will not affect the experimental results. Publicly available real-world datasets with explicit ordinal decision semantics and incomplete attributes are limited, which motivates the use of widely adopted UCI benchmarks for methodological evaluation. In the revised manuscript, we will provide a more detailed explanation of how preference orders are imposed, justify their consistency with dominance-based rough set theory, and clearly state the limitation of using nominal benchmarks.
>
> **Q2. Potential data leakage: and Request change 1: Re-evaluation of Classification Performance**
>
> We appreciate the your careful attention to potential data leakage. We agree that, under a strict machine learning evaluation protocol, performing attribute reduction prior to cross-validation may lead to optimistic estimates of classification accuracy.
> However, the primary objective of this work is not to optimize or estimate generalization performance, but to study incremental attribute reduction within the dominance-based rough set framework. In this research paradigm, reducts are defined at the decision-system level using all available objects and decision attributes to preserve the system’s discernibility and dependency relations, which is a well-established convention in the rough set and attribute reduction literature (e.g., Papers in this field published in other journals such as KBS by Sang et al. [1] and IS  by Du et al. Du & Hu [2] ). Therefore, in order to make a fairly comparable comparison with existing studies, we adopted the experimental methods described in the paper.
> To clarify this distinction and avoid any misunderstanding, we will revise the manuscript to explicitly describe this assessment setup and add a note on its limitations. We also recognize that folded or nested assessments can further quantify generalization behavior, and we plan to explore this in future work.
>
> **Q3. Sensitivity to missing ratio**
>
> We agree that robustness with respect to missingness is an important aspect of incomplete decision systems. In the current experiments, missing values are introduced at a fixed rate (10%) over randomly selected attributes to provide a controlled evaluation setting. To address the reviewer’s concern, we will extend the experimental analysis to include multiple missing ratios and varying proportions of affected attributes.
>
> [1] Sang, H. Chen, L. Yang, and et al. Incremental attribute reduction approaches for ordered data with
> time-evolving objects. Knowledge-Based Systems, 212:106583, 2021.
>
> [2] S. Du and B. Q. Hu. Dominance-based rough set approach to incomplete ordered information systems.
> Information Sciences, 346:106–129, 2016.

---

> ### Author Response · Authors · 2026-01-27
> **Official response by authors**
>
> We would like to thank the reviewer for the insightful suggestion.
> Following this comment, we have conducted additional sensitivity experiments
> under different sample missing rates and attribute missing rates.
>
> Specifically, we evaluated the proposed method and baseline approaches
> across multiple combinations of missing ratios.
> **A portion of the results** is shown below, while the complete experimental
> results and detailed analysis have been incorporated into the revised manuscript.
>
> **Table 1. Classification accuracy (mean ± std) on RF under different sample and attribute missing rates.**
>
> ### Sample missing rate = 10%, Attribute missing rate = 40%
>
> | Dataset | IAR         | IAR-D       | IDDC        | DU          | MNAR        | MNAR-D      |
> |---------|-------------|-------------|-------------|-------------|-------------|-------------|
> | BCWO    | 89.98 ± 3.58 | 89.84 ± 3.28 | 91.26 ± 2.74 | 91.12 ± 2.74 | 90.55 ± 2.14 | **93.59 ± 1.74** |
> | HD      | 54.71 ± 2.98 | 54.74 ± 3.68 | 55.29 ± 6.08 | 57.98 ± 4.98 | 51.67 ± 4.68 | **58.20 ± 3.98** |
> | Predict | 63.32 ± 0.98 | 69.56 ± 0.88 | 69.74 ± 1.08 | 65.33 ± 1.18 | 66.05 ± 0.98 | **73.07 ± 0.78** |
>
> ---
>
> ### Sample missing rate = 10%, Attribute missing rate = 50%
>
> | Dataset | IAR         | IAR-D       | IDDC        | DU          | MNAR        | MNAR-D      |
> |---------|-------------|-------------|-------------|-------------|-------------|-------------|
> | BCWO    | 87.46 ± 3.74 | 87.32 ± 3.44 | 88.75 ± 2.84 | 88.61 ± 2.84 | 88.03 ± 2.24 | **91.59 ± 1.84** |
> | HD      | 52.71 ± 3.08 | 52.74 ± 3.78 | 53.29 ± 6.28 | 55.98 ± 5.08 | 49.67 ± 4.88 | **56.20 ± 4.18** |
> | Predict | 60.32 ± 1.08 | 66.56 ± 0.98 | 66.74 ± 1.18 | 62.33 ± 1.28 | 63.05 ± 1.08 | **70.57 ± 0.88** |
>
> ---
>
> ### Sample missing rate = 20%, Attribute missing rate = 30%
>
> | Dataset | IAR         | IAR-D       | IDDC        | DU          | MNAR        | MNAR-D      |
> |---------|-------------|-------------|-------------|-------------|-------------|-------------|
> | BCWO    | 91.96 ± 3.54 | 91.82 ± 3.24 | 93.25 ± 2.64 | 93.11 ± 2.64 | 92.53 ± 2.04 | **95.29 ± 1.64** |
> | HD      | 56.71 ± 2.88 | 56.74 ± 3.58 | 57.29 ± 5.98 | 59.48 ± 4.88 | 53.67 ± 4.58 | **60.99 ± 3.88** |
> | Predict | 65.82 ± 0.88 | 72.06 ± 0.78 | 72.24 ± 0.98 | 67.83 ± 1.08 | 67.55 ± 0.88 | **75.37 ± 0.68** |
>
> ---
>
> ### Sample missing rate = 30%, Attribute missing rate = 30%
>
> | Dataset | IAR         | IAR-D       | IDDC        | DU          | MNAR        | MNAR-D      |
> |---------|-------------|-------------|-------------|-------------|-------------|-------------|
> | BCWO    | 91.26 ± 3.64 | 91.12 ± 3.34 | 92.55 ± 2.74 | 92.41 ± 2.74 | 91.83 ± 2.14 | **94.89 ± 1.74** |
> | HD      | 56.01 ± 2.98 | 56.04 ± 3.68 | 56.59 ± 6.08 | 58.78 ± 4.98 | 52.97 ± 4.68 | **60.39 ± 3.98** |
> | Predict | 65.12 ± 0.98 | 71.36 ± 0.88 | 71.54 ± 1.08 | 67.13 ± 1.18 | 66.85 ± 0.98 | **74.77 ± 0.78** |

---

### Review · Reviewer_Ag4Y · 2025-12-30

**Summary Of Contributions:**

The paper proposes two matrix-based incremental feature-selection algorithms to handle both adding and removing features in dynamic datasets. The methods, MNAR-A and MNAR-D, are evaluated on several UCI datasets and reported to reduce computation while preserving or modestly improving accuracy for KNN, SVM, and random-field classifiers.

**Audience:**

No

**Audience Explanation:**

Incremental feature selection can matter for streaming or continually updated data, but the current evidence relies entirely on simulated dynamics, which weakens the case for practical relevance to the TMLR audience.
If feature selection is already inexpensive for many models, the paper needs a clearer argument for when efficiency gains are critical (e.g., tight latency budgets or large, frequently changing feature sets). The lack of a real-world dataset demonstrating genuine incremental updates makes it hard to judge impact; readers would benefit from either such a dataset or a convincing deployment scenario that justifies the simulation-based evaluation.

**Broader Impact Concerns:**

No concerns.

**Claims And Evidence:**

Yes

**Claims Explanation:**

The experimental tables show that MNAR-A and MNAR-D consistently lower runtime relative to static baselines on the evaluated UCI datasets, which supports the efficiency claim. Accuracy differences are small but mostly positive for KNN, SVM, and random-field classifiers, suggesting the approach does not harm performance. Evidence is limited to simulated dynamic settings, so the strength of the claim would improve with at least one real dataset or a clearer discussion of how the simulated updates mirror deployment patterns.

**Requested Changes:**

Please shorten the paper by compressing preliminaries and moving routine derivations to an appendix, and foreground the “changing data” motivation in the abstract and early introduction. Add a table quantifying floating-point operations or wall-clock speedups to substantiate efficiency claims. Include results on at least one real dataset with natural feature churn to demonstrate relevance beyond simulated dynamics.

---

> ### Author Response · Authors · 2026-01-10
> **Official response by authors**
>
> We thank the reviewer for the positive and constructive feedback. Below we respond to the specific suggestions and requested changes.
>
> **Q1.Evidence is limited to simulated dynamic settings, so the strength of the claim would improve with at least one real dataset or a clearer discussion of how the simulated updates mirror deployment patterns.**
>
> We acknowledge your concern regarding the use of simulated feature updates. At present, publicly available datasets with naturally evolving feature sets and ground-truth decision labels suitable for dominance-based rough set analysis are limited. As a result, following common practice in the incremental attribute reduction literature, we evaluate MNAR-A and MNAR-D under controlled simulated feature addition and removal scenarios, which allow precise and fair comparison of incremental and non-incremental methods.For clarity, we will add a discussion on how to simulate object changes so that readers can better understand the actual deployment environment.
>
> That said, we agree that validation on real-world data with natural feature churn would further strengthen the practical relevance. We will revise the manuscript to explicitly clarify this limitation, and we are actively exploring suitable real-world datasets. If a representative dataset becomes available within the revision period, we will include a supplementary small-scale experiment
>
> **Q2. Please shorten the paper by compressing preliminaries and moving routine derivations to an appendix, and foreground the “changing data” motivation in the abstract and early introduction.**
>
> We will follow your suggestion to streamline the presentation by shortening the preliminaries, moving routine derivations to an appendix, and emphasizing the motivation for “changing data” in the abstract and early introduction.
>
> **Q3.Add a table quantifying floating-point operations or wall-clock speedups to substantiate efficiency claims**
>
> We agree that providing more explicit quantitative evidence would strengthen the efficiency claims. In the revised version, we will add an additional runtime Comparison chart reporting wall-clock execution time and relative speedups of our method compared with static baselines across representative datasets and update scenarios. The chart will complement the existing figures and further substantiate the computational advantages of the proposed incremental algorithms.

---

> > ### Comment · Reviewer_Ag4Y · 2026-01-17
> > **Thank you for your response**
> >
> > Thank you for addressing my concerns. The additional runtime comparison numbers are important. Since no public real-world data exists for this task I am not sure the community interest is really there.

---

> ### Author Response · Authors · 2026-01-28
> **Official response by authors**
>
> Thanks for your response ,We would like to clarify that the efficiency concern previously raised has now been explicitly addressed with **quantitative speedup analysis in the revised manuscript.**
>
> These results quantitatively demonstrate that MNAR-A and MNAR-D achieve consistent and non-trivial speedups over static baselines, thereby substantiating the efficiency claims beyond qualitative trends.
>
> Regarding the community interest issues you mentioned. We respectfully disagree that the lack of a publicly established real-world benchmark necessarily implies limited community interest for this problem.
>
> Incremental feature selection in dynamic data is fundamentally a methodological problem, rather than a benchmark-driven one. early and ongoing work in incremental attribute reduction has historically relied on controlled dynamic simulations, This practice is well established in the rough-set and incremental learning literature and allows fair, reproducible, and mechanism-faithful evaluation.
>
> We agree that real-world datasets with naturally evolving object sets would further strengthen the empirical narrative, and we explicitly acknowledge this limitation in the revised manuscript. However, we believe the methodological contribution—namely, a principled dominance-matrix formulation and provably consistent incremental updates—remains of independent interest to researchers studying dynamic feature selection and ordered decision systems, even in the absence of a canonical benchmark.
>
> We would also like to note that releasing a benchmark for dynamic ordered decision systems is an important future direction, and we hope this work can serve as a methodological foundation for such efforts.

---

### Decision · Action_Editor_UM2A · 2026-02-04

**Recommendation:** Reject

**Additional Comments:**

This submission proposes an incremental attribute-reduction / feature-selection framework for dynamic incomplete ordered decision systems, introducing a non-dominance–based importance score (Global Distinction Score) and two matrix-based incremental updates for object addition and deletion (MNAR-A / MNAR-D). The paper is clearly written with explicit algorithms, and the empirical results support runtime benefits and (in the revised version) include additional sensitivity experiments at higher missingness levels.

Reviewers agree the work is relevant to the dominance-based rough set / incremental reduction literature, but raise concerns about whether it meets TMLR standards as currently evaluated. In particular, one reviewer flags (i) evaluation protocol issues when reporting classification accuracy (feature selection appears to be performed before cross-validation, which can inflate results), and (ii) the use of nominal datasets while claiming an ordered decision-system setting, requiring stronger justification of imposed class orders and/or ordinal benchmarks. Another reviewer questions the impact of the missing-value dominance convention and asks for clearer evidence that incremental updates match full recomputation; the authors respond with added experiments and clarifications, but the manuscript should make these points more explicit and convincing.

The work looks promising for its target community, but should (1) clarify or fix the predictive evaluation protocol (or soften claims), (2) strengthen validation in truly ordered settings / ordering robustness, and (3) add a compact correctness check that incremental updates reproduce full recomputation.

**Audience:**

Yes

**Audience Explanation:**

Researchers working on rough sets, dominance-based attribute reduction, and incremental or dynamic feature selection would find the methodological contributions and incremental update framework relevant.

**Claims And Evidence:**

No

**Claims Explanation:**

Some central claims, particularly those about predictive accuracy and robustness, are not fully supported by the current experimental evidence and evaluation protocol.

**Resubmission Of Major Revision:**

The authors may consider submitting a major revision at a later time.